

# High ice water content in tropical mesoscale convective systems (a conceptual model)

Alexei Korolev[1], Zhipeng Qu[1], Jason Milbrandt[1], Ivan Heckman[1], Mélissa Cholette[1], Mengistu Wolde[2], Cuong Nguyen[2], Greg M. McFarquhar[3], Paul Lawson[4], Ann M. Fridlind[5]

[1] Environment and Climate Change Canada, Toronto, ON, Canada
[2] National Research Council Canada, Ottawa, ON, Canada
[3] Cooperative Institute for Severe and High Impact Weather Research and Operations and School of Meteorology, University of Oklahoma, Oklahoma City, OK, USA
[4] Stratton Park Engineering Company, Boulder, CO, USA
[5] NASA Goddard Institute for Space Studies, New York, NY, USA

*Correspondence to*: Alexei Korolev (alexei.korolev@ec.gc.ca)

**Abstract**
The phenomenon of high ice water content (HIWC) occurs in mesoscale convective systems (MCS) when a large number of small ice particles with typical sizes of a few hundred micrometers, concentrations of the order of $10^2$-$10^3$ L$^{-1}$ and IWC exceeding 1g m$^{-3}$ are found at high altitudes. HIWC regions in MCSs may extend vertically up to 10 km above the melting layer and horizontally up to hundreds of kilometers, filling large volumes of the convective systems. HIWC has great geophysical significance due to its effect on precipitation formation, the hydrological cycle, and the radiative properties of MCSs. It is also recognized as a hazard for commercial aviation operations since it can result in engine power loss and in the malfunctioning of aircraft data probes. This study summarizes observational and numerical simulation efforts leading to the development of a conceptual model for the production of HIWC in tropical MCSs based on the data collected during the HAIC-HIWC campaign in French Guiana in 2015. It is hypothesized that secondary ice production (SIP) in the vicinity of the melting layer plays a key role in the formation and sustainability of HIWC. In-situ observations suggest that the major SIP mechanism in the vicinity of the melting layer is related to the fragmentation of freezing drops (FFD). Both in-situ data and numerical simulations suggest that the recirculation of drops through the melting layer could lead to the amplification of SIP. However, laboratory measurements remain insufficient to support the accurate model representation of FFD conclusively. The proposed conceptual model and simulation results motivate further efforts to extend reproducible laboratory measurements.

## 1. Introduction

The interest in cloud environments with enhanced ice water contents (IWC > 1 g m$^{-3}$) has emerged in connection with the development of aviation safety envelopes for the operation of commercial aviation. Reports of a growing number of engine power-loss events on commercial aircraft flying in the vicinity of convective storm cores has initiated trial field studies of the environmental conditions related to the engine events (Lawson et al., 1998; Strapp et al., 1999).



Data obtained provided evidence that engine power-loss events result primarily from the
ingestion of large concentrations of ice crystals into the engine. The analysis of the Boeing
event database showed that the engine power losses typically occur in tropical mesoscale
convective systems (MCS) in the altitude range of 5 to 12 km and the temperature range of -
10°C to -55°C (Mason et al., 2006; Mason and Grzych, 2011). The radar reflectivity in the
regions of engine events was found to be typically below 30 dBZ (e.g., Mason et al., 2006;
Grzych and Mason, 2010; Protat et al., 2016; Wolde et al., 2016). A detailed examination of the
information obtained allowed the formulation of a hypothesis that the engine power-loss
events occur in the presence of high concentrations of small ice crystals constituting high ice
water content (HIWC) environments (Lawson et al., 1998; Strapp et al., 2016).

In order to provide a statistically significant data set to assess aviation safety envelopes for
engine operations in HIWC environments (FAA Title 14 Code of Federal Regulations Part 33
Appendix D – Mixed-Phase/Glaciated Environmental Envelope, and the identical EASA CS-25
Appendix P), the USA Federal Aviation Administration (FAA) and European Union Aviation
Safety Agency (EASA) jointly initiated an international field campaign: the European High-
Altitude Ice Crystals (HAIC) (Dezitter et al., 2013) and North American HIWC (hereafter HAIC-
HIWC) (Strapp et al., 2016). The flight operations were conducted out of Darwin, Australia in
2014 and Cayenne, French Guiana in 2015. The HAIC-HIWC project was designed to obtain the
99[th] percentile of total water content (TWC) and characterize ice particle sizes for regulatory
purposes. To address the regulatory objectives, cloud environments were sampled along the
horizontal traverses at different distances from the convective cores at four temperature levels
centered around -10°C, -30°C, -40°C and -50°C (±5°C) (Strapp et al., 2020).

During the HAIC-HIWC in Darwin-2014 and Cayenne-2015 campaigns, most of the MCSs
were sampled during its mature stage, i.e., when the cloud top area with temperature below -
50°C from the GOES-13 10.8 $\mu$m channel passed its maximum. Approximately, 83% of studied
MCSs had an age of over 6 h, and approximately 44% over 12 h (Hu et al., 2021). Most of the
MCSs were in ice phase thermodynamic states at the moment of their observation. The spatial
occurrence of mixed-phase cloud segments was only a few percent (Korolev et al., 2018, Strapp
et al., 2021).

The first two HAIC-HIWC flight campaigns provided a wealth of data which enabled insight
into the microphysical and thermodynamical conditions inside tropical MSCs. Thus, Leroy et al.
(2017) found that in HIWC cloud regions the median mass diameter (MMD) of ice particles
ranges from 250 $\mu$m to 500 $\mu$m and that the MMD decreases with increasing IWC. This is
consistent with earlier findings of Lawson et al. (1998), who showed that median of the
maximum particle dimension in tropical MCSs is close to 400 $\mu$m. Strapp et al. (2020, 2021)
studied the dependence of the TWC versus averaging spatial scale. It was found that maximum
TWC reached 4.1 g m[-3] at averaging distance 0.5 nm (0.93 km) at the -30°C temperature level.
Even at the 100 nm (186 km) distance scale, the overall TWC maxima were near 2 g m[-3] in the
−10°C and −30°C levels. Average MMD values increased with increasing temperature from
about 326 μm in the −50°C level to about 708 μm in the −10°C level (Strapp et al. 2020).

Ladino et al. (2017) concluded that the measured concentration of ice particles could not be
explained by primary ice nucleation and that the likely explanation of the formation of the
observed ice was related to secondary ice production (SIP). Analysis of high-resolution imagery
of ice particles performed in Korolev et al. (2020) suggested that SIP occurs above the melting



layer primarily due to the fragmentation of freezing drops SIP mechanism. Most of the observed SIP cloud regions were collocated with convective updrafts.

Hu et al. (2021) explored dependences of IWC, MMD and ice number concentration $N_{ice}$ versus environmental conditions such as temperature ($T$), vertical velocity ($U_z$), proximity to the convective core ($L_{conv}$) and MSC age. It was found that IWC correlates with $U_z$, whereas MMD decreases and $N_{ice}$ increases with decreasing $T$. The relationship between IWC and MMD is more complex and it depends on $T$, $N_{ice}$, $L_{conv}$, and MCS age.

Hu et al. (2022) and Brechner et al. (2023) explored the modality of ice particle size distributions (PSD) in HIWC cloud regions, developed functional relationships between the PSD parameters, and examined the dependence of PSDs on environmental conditions. Using an automated technique for the identification of unimodal, bimodal, and trimodal size distributions, they found that temperature and IWC were the most important variables
determining the modality of the PSDs, with the frequency of trimodal distributions increasing with temperature and bimodal distributions more common in updraft cores, and the existence of the small peak indicative of SIP.

Early numerical simulations of the HIWC phenomenon showed significant challenges in reproduction of HIWC environment and revealed great sensitivity to the ice initiation schemes
employed in numerical models (Ackerman et al., 2015; Fridlind et al., 2015; Franklin et al., 2016; Stanford et al., 2017; Qu et al., 2018; Huang et al., 2021). To better simulate the HIWC condition, Wurtz et al. (2021, 2023) improved the parameterization of snow particle size distributions within a single-moment cloud microphysical scheme.  Qu et al. (2018) used a cloud-resolving (250-m horizontal grid spacing) model to simulate tropical MCS with SIP represented by the Hallett-
Mossop (rime splintering) SIP process. It was found that the that the misaligned simulated profiles of $IWC$ and $N_{ice}$ might be related to inadequate handling of SIP processes which led to the production of too few, and thus too large, ice particles. It was thus recognized that the proper simulation of hydrometeors in HIWC environments could be improved with a better representation of SIP. The subsequent development of numerical models showed how inclusion
of SIP processes plays a crucial role in the simulation of HIWC environment in tropical MCSs (Qu et al., 2022; Huang et al., 2021, 2022).

The geophysical significance of HIWC environments in tropical MCSs has yet to be fully understood. However, it is expected that ice initiation mechanisms in the convective cores of tropical MCSs are directly linked to the formation of anvil cirrus and their longevity, which
eventually affects radiative transfer. The typical size of ice particles constituting HIWC environments is strongly related to their fall velocity, which affects the sustainability of HIWC cloud regions and, thus, will affect the type (e.g., hail vs. rain) and the rate of precipitation at the surface. Altogether, this suggests that the microphysical processes in tropical MCSs may affect climate, circulation, and the hydrological cycle on regional and global scales (Sullivan and
Voigt, 2021).

As indicated above, the HIWC environment in tropical MCSs typically has radar reflectivity values below 30 dBZ (e.g., Mason et al., 2006; Grzych and Mason, 2010; Protat et al., 2016; Wolde et al., 2016), which corresponds to green echoes on the pilot radar. This hinders in-flight identification of regions of HIWC and the initiation of an avoidance procedure. Numerical
guidance from high-resolution weather prediction models is a potential tool to help establish avoidance strategies for rerouting flights and mitigating negative impacts of encounters with HIWC. To this end, a conceptual model underlying the natural formation of HIWC environments



and in-depth understanding of the cloud microphysical and thermodynamical processes in MCSs should precede the development of such numerical guidance.

The objective of this paper is to synergize observational and modeling efforts and formulate a preliminary conceptual model of the formation of sustainable HIWC regions in tropical MCSs. The in-situ airborne observations are essentially Eulerian type measurements, which suffer from problems such as small sample volume and sparse needle-like sampling (Baumgardner et al., 2017) as well as the difficulty of gaining information on processes since air parcels are not tracked. SIP processes occur on short time periods (minutes) and small spatial scales (hundreds of meters) (Korolev et al., 2020). Therefore, in-situ observations of SIP events inside mesoscale cloud systems are a great challenge, and sampling SIP is a matter of luck. On the other hand, in-situ observations give numerous hints on cloud processes and provide feedback for the improvement of numerical simulations. Upon improvement, numerical models can potentially help to put together bits of information collected from in situ and fill the gaps in our understanding, thereby giving a deeper insight of the microphysics and thermodynamics of tropical MCSs. We have attempted to employ this iterative process between analysis of observations and numerical simulations, aiming at the formulation of a conceptual model of HIWC in tropical MCSs.

The present paper is arranged as follows. A summary of the results of the in-situ observations of HIWC microphysics are presented in section 2. Section 3 presents results of quasi-idealized cloud-resolving numerical simulations of HIWC in tropical deep convection. A proposed conceptual model of HIWC formation is discussed in section 4 followed by conclusions in section 5.

## 2. Results of in-situ observations

In this section, we describe the results of the in-situ observations of HIWC collected during the HAIC-HIWC campaign, which are complimentary to the previous studies above.

### 2.1 Methodology, instrumentation and the data set

In the absence of a clear physical basis, at present there is no consensus regarding the definition of HIWC. Typically, the threshold IWC adapted by different research groups varies between 2 g m$^{-3}$ (Leroy et al., 2016), 1.5 g m$^{-3}$ (Leroy et al. 2017; Hu et al., 2021), and 1 g m$^{-3}$ (Yost et al., 2018; Strapp et al., 2020, 2021). In this study, we will employ the threshold of 1 g m$^{-3}$ to define HIWC.

The in-situ observations presented here are primarily focused on the data collected from the National Research Council of Canada (NRC) Convair-580 (CV580) research aircraft during the HAIC-HIWC project. The flight operations were conducted out of Cayenne in May 2015. Fourteen CV580 research flights were conducted in the frame of the HIWC campaign, with an average flight endurance of approximately 4 h. Most of the flights were performed in oceanic MCS at altitudes ranging from 4700 m to 7300 m and temperatures from 0°C to -15°C. The observations of MCSs were performed during their mature stages when the area of clouds with longwave brightness temperatures less than -50°C from GOES-13 approached or surpassed its maximum. At that stage, most of the volume of the MCS above the freezing level was nearly glaciated, with embedded mixed-phase regions associated mainly with updrafts (Korolev et al., 2018; Strapp et al., 2020). However, the MCSs studied remained dynamically active during the observation periods, with updrafts peaking at 15-20 m s$^{-1}$. The CV580 was equipped with state-of-the-art cloud microphysical and thermodynamic instrumentation. Size distributions of



aerosol particles were measured by a DMT Ultrahigh Sensitivity Aerosol Spectrometer (UHSAS) (Cai et al., 2008). Measurements of $N_{ice}$ and IWC were extracted from composite PSDs

measured by optical array 2D imaging probes, such as PMS Optical Array Probe (OAP) OAP-2DC (Knollenberg, 1981), a SPEC 2-Dimensional Stereo (2DS, Lawson et al., 2006) and a DMT Precipitation Imaging Probe PIP (Baumgardner et al., 2001). Cloud droplet size distributions were measured by a PMS Forward Scattering Spectrometer Probe (FSSP, Knollenberg, 1981) and a DMT Cloud Droplet Probe (CDP-2, Lance et al., 2010). The SPEC Cloud Particle Imager

(CPI, Lawson et al., 2001) provided a photographic quality 256 grey-level imagery of cloud particles with 2.3 μm pixel resolution. Bulk liquid water content (LWC) and TWC were measured with a SkyPhysTech Nevzorov probe (Korolev et al., 1998) and a SEA IsoKinetic probe (IKP-2) (Davison et al., 2011). A Goodrich Rosemount Icing Detector (RID) was used for detection of liquid water at $T$<-5°C (Mazin et al., 2001). The extinction coefficient was measured with the

ECCC Cloud Extinction Probe (Korolev et al., 2014). Vertical velocity was measured by Rosemount 858 (Williams and Marcotte, 2000) and an Aventech AIMMS-20 (Beswick et al., 2008). Air temperature was measured by a Rosemount total-air temperature probe and an AIMMS-20. Water vapor mixing ratio was measured by Licor-850a and Licor-6262. The CV580 was also equipped with NRC Airborne W-band and X-band radars (NAWX) with Doppler

capability (Wolde and Pazmany, 2005).

Measurements of $N_{ice}$ remain a challenging task primarily due to the ambiguity of the definition of the size of non-spherical particles (e.g., McFarquhar et al., 2017; Korolev et al., 2017), the uncertainty of the definition of sample area (e.g., Baumdargner et al., 2017; McFarquhar et al. 2017), and artifacts related to diffraction effects (e.g., Korolev and Field,

2015) and ice particles shattering during sampling (e.g., Korolev et al., 2011, 2013b). These effects are most pronounced for particle images consisting of less than four pixels in size, regardless of image pixel resolution of the particle probe. For that reason, particle images with a size of less than four pixels were excluded from the analysis of 2DS data. Therefore, the in situ data describing ice microphysical parameters below refer to particles ≥ 40 μm.

To mitigate the effects of shattering artifacts anti-shattering K-tips were installed on all particle probes (Korolev et al., 2013a). The remaining shattering artifacts were filtered out with the modified inter-arrival time algorithm (Korolev and Field, 2015). Diffraction effects resulted in diffraction fringes, noisy patterns around images and fragmented images when particles passed near the depth-of-field. These were cleared with special image processing algorithms.

The collected cloud microphysical and remote sensing data were processed and analyzed with the ECCC D2G software.

Altogether, the NRC CV580 instrumentation provided high-level redundant measurements and allowed extensive analysis of cloud microphysics and cloud dynamics.

During the HAIC-HIWC field campaign the measurements were also performed from the

SAFIRE Falcon-20 (F20) (Leroy et al., 2016; Strapp et al., 2021). Because of the limited access, the SAFIRE F20 data could not be processed the same way as the NRC CV580 measurements. For this reason, the observations used in this study is primarily focused on the CV580 data. However, published SAFIRE F20 results are employed to support the outcomes of this study.

**2.2 Ice initiation**

The mechanism behind the ice initiation is the key to understanding the formation of HIWC in tropical MCSs. One of the most striking findings of this research was the observation of



unusually high amounts of small facetted ice crystals with sizes < 100 μm formed just above the melting layer at temperatures -5°C $< T <$ -2°C (Korolev et al., 2020). Typically, the observed

small ice crystals were hexagonal solid prisms with aspect ratios ranging from 0.3 to 6. Examples of such small ice crystals are shown in Fig. 1a. The concentration of small ice crystals ranged from $10^2$ to $10^3$ L$^{-1}$, which exceeds the estimated concentration of INPs ($N_{INP}$, Ladino et al., 2017) by 6-8 orders of magnitude. Such a big difference between $N_{ice}$ and $N_{INP}$ suggests that the initiation of small facetted ice particles is related to SIP (Field et al., 2017).

At present, there are six recognized SIP mechanisms (Korolev and Leisner, 2020). Identification of the specific SIP mechanisms responsible for the formation of the observed ice is hindered by the essentially Eulerian nature of the observations. Such observations do not allow direct observations of SIP processes, but rather enable the observation of the products of SIP after the SIP events have occurred. In this regard, the analysis of the cloudy environment

associated with SIP may provide some hints about the mechanisms responsible for the enhanced ice concentration. The important condition for using of such evidence is that they be spatially correlated with the SIP environment.

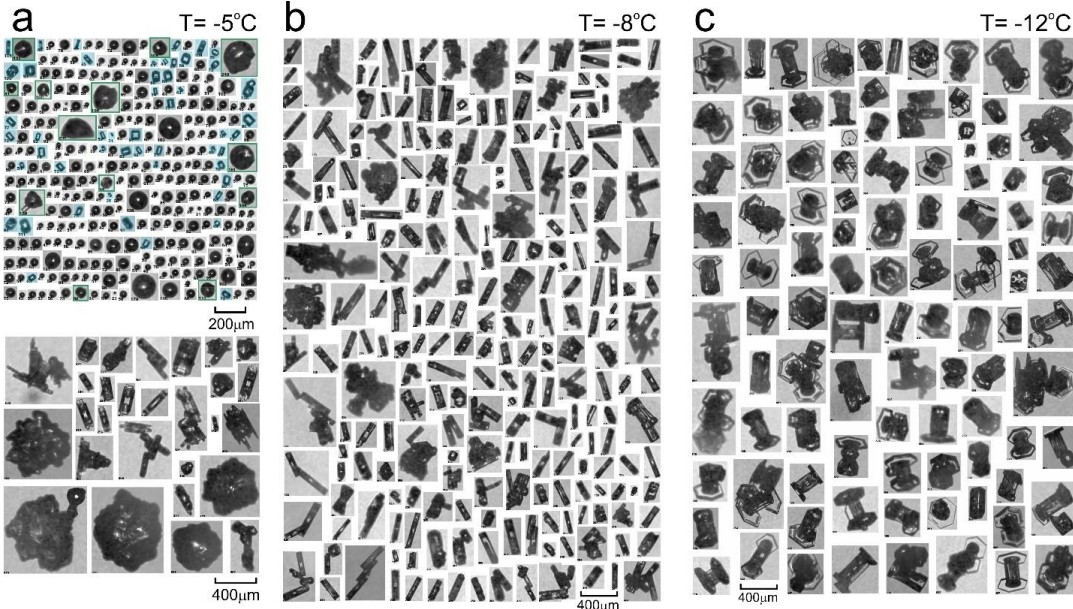

**Figure 1**. Images of ice particle observed in MCS during the NRC Convair-580 flight on 15 May 2015 measured by

the SPEC CPI. (a-top): spatial sequence of liquid drops and small facetted ice crystals (highlighted by blue), deformed and fragmented frozen drops (green frames), (a-bottom): large ice particles from the SIP region shown in a-top 09:40:42–09:40:47, 5600m, $T$=-5.1°C, $IWC \approx$0.6 g m$^{-3}$, $N_{ice} \approx$ 1000 L$^{-1}$; (b) spatial sequence of ice particles measured in the cloud region dominated by columnar ice, 10:49:58-10:50:04, 6250m, $T$=-7.8°C, $IWC \sim$1.8 g m$^{-3}$, $N_{ice} \approx$ 600 L$^{-1}$; (c) spatial sequence of particles in the region dominated by capped columns 11:47:38-11:47:50,

7000 m, $T$= -11.7°C, $IWC \sim$0.9 g m$^{-3}$, $N_{ice} \approx$ 100 L$^{-1}$.

The age of the small facetted ice crystals at -5°C is estimated as less than 2-3 min (Korolev et al. 2020). As argued in the aforementioned study, the point of origin of such small-age ice particles is near the location of their observation. This enables the use of the evidence found in

the environment populated by small facetted ice crystals for identification of the possible type of the SIP mechanism. Thus, it was found that in many cases the regions with enhanced



concentrations of small facetted ice were associated with the presence of supercooled large drops (SLD) and they also contained fragments of deformed and fragmented frozen droplets. Such droplets are highlighted by green frames in Fig. 1a (see also Fig. 18 in Korolev et al., 2020).

Observations of precipitation size drops, fragmented and deformed frozen drops, and large concentrations of tiny facetted ice crystals suggest that the "fragmentation of freezing droplet" (FFD) SIP mechanism is responsible for the enhanced concentration of the observed ice concentration above the melting layer.

## 2.3  Metamorphosis of ice particle shapes

Early laboratory studies revealed that the linear growth rate ($dL/dt$) of ice particles along the crystallographic axes (i.e., $a$ and $c$) is highly sensitive to humidity and ambient temperature (e.g., aufm Kampe et al., 1951; Nakaya, 1954; Kobayashi, 1957; Hallett and Mason, 1958). This unique feature of ice growth results in an infinite variety of intricate ice particles shapes.

However, despite the complexity and variety of ice shapes, there are two major growth regimes. The first occurs, when the growth rate of the prism faces (i.e., along the $c$-axis) exceeds the growth rate of the basal faces (i.e., along $a$-axes) e.g., $dL_c/dt > dL_a/dt$ . This regime results in a formation of columnar type of ice crystals, e.g., solid and hollow columns, scroll, sheath, needles and bullets. The second growth regime corresponds to the case when

the growth rate along the $c$-axis is slower compared to the growth rate along $a$-axes, e.g., $dL_c/dt < dL_a/dt$ . This regime corresponds to the formation of planar types of ice crystals such as plates, stellar, and dendrites. The preferential growth directions of ice crystals along $a$- and $c$-axes change versus temperature in a cyclical way, changing the habits between plate-column-plate-column near -4°C, -9°C and -22°C, respectively. Temperature ($T$) and relative

humidity ($RH$) experience continuous changes in a cloud environment due to the air vertical motion and mixing. This results in metamorphoses of ice particle shapes due to changes of the dominant growth direction, depending on $T$ and $RH$ at each moment of time. Therefore, in some special cases, observations of ice particle habits may allow the reconstruction of the history of $T$ and $RH$, which the ice particles had experienced. In this section, we consider ice

particle habits observed in tropical MCSs, which provide information on ice particle history and their vertical travel.

One of the important features of tropical MCSs at temperatures -10°C < $T$ <-5°C is that, in many cases, columnar ice is the dominant habit of particles constituting HIWC regions. The fraction of columns in HIWC regions in some cases may approach 100%. An example of ice

particle images in such HIWC environment is shown in Fig. 1b.

The typical size of the observed columns varied from 200 to 500 μm and their concentration changes in the range from $10^2$ to $10^3$ L$^{-1}$. This is quite a high concentration of ice and it cannot be explained by primary ice nucleation in the -10°C < $T$ < -5°C temperature range (Ladino et al. 2017). The similarity of the concentrations of columns at -10°C <$T$< -5°C and small facetted

secondary ice crystals at -5°C <$T$< -2°C, and the spatial proximity in the vertical direction of these two temperature layers is indicative that the columnar ice in the HIWC cloud region is consistent with the successive growth of the secondary ice formed in the vicinity of the melting layer after their ascent in convective updrafts.

Further up in the cloud, at the levels with $T$< -12°C, it was found that the population of ice

crystals in HIWC regions contained a large number of capped columns. Similar to columnar ice, in some cases the fraction of capped columns may reach nearly 100%, as shown in Fig. 1c.



Observations of capped columns in tropical MCSs were also reported by Ackermann et al. (2015, Fig. 1). Leroy et al. (2017, Fig. 8) presented measurements of ice particles images dominated by capped columns, which were sampled at -37°C in the HIWC cloud region during
the HAIC-HIWC campaign.

Capped columns are a result of the metamorphosis of columnar ice crystals after transitioning from the columnar growth regime to the plate growth regime determined by the ambient $T$ and $RH$. In this case, the pre-existing columnar crystals alter their primary growth direction from the $c$-axis direction to the $a$-axes. However, due to higher gradients of water
vapor around sharp ice edges the crystal areas in the vicinity of the edges of the basal face will have a higher growth rate along the $a$-axes compared to that of the rest of the prismatic faces (e.g., Nelson, 2001). This leads to the formation of plates capping the ice column ends.

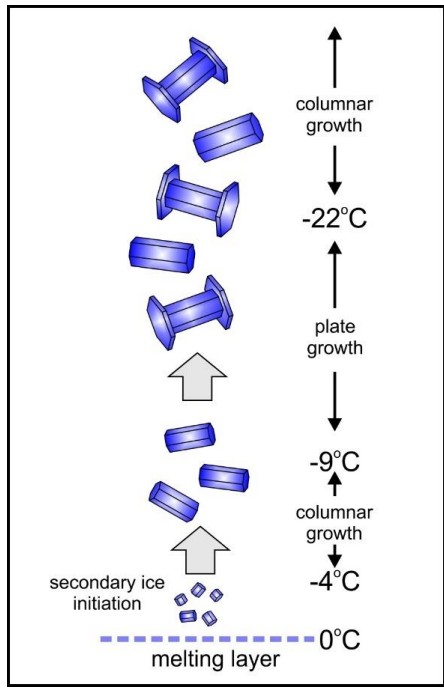


**Figure 2**. A conceptual diagram of metamorphosis of ice particles in convective regions in tropical MCSs.

The association of columns and capped columns with HIWC regions and their linkage through the metamorphosis of the ice shapes suggests that the capped columns originate from
the columns formed at -10°C < $T$ < -5°C. Formation of capped columns may also be associated with the metamorphosis of columns formed at high altitudes at $T$ < -22°C that then precipitated down to the levels with temperatures -9°C > $T$ > -22°C, corresponding to the plate growth condition. Such a scenario was discussed by Heymsfield et al. (2002). This case may be also relevant to some fraction of ice crystals in the studied MCSs. However, such sequence of ice
transformation cannot explain observations of capped columns at temperatures of -37°C (Leroy et al., 2017) since it assumes that the original columns were formed at lower temperatures at a



higher altitude and upon precipitating down to the -37°C level they cannot experience the plate growth conditions; therefore, they cannot transform into capped columns.

Observations of capped columns at temperatures around -37°C can be also explained by
recirculation through the levels with -9°C> $T$ >-22°C of the columns formed at $T$ <-22°C. However, such a scenario seems unlikely to explain nearly uniform population ice crystals consisting of columns and capped with large spatial extension and concentration reaching a few hundred L$^{-1}$ (Leroy et al., 2017).

Summarizing the above, the most likely scenario for the evolution of ice particles in the
convective cloud regions in tropical MCSs is as follows. Secondary ice initiation occurs in the vicinity of the melting layer in the temperature range -5°C < $T$ < -2°C, resulting in the generation of numerous small facetted ice crystals. After ascent within the convective updraft to the level -9°C< $T$ < -5°C the faceted small ice crystals turn into columns. Subsequent ascent though the temperature range -22°C < $T$ <-9°C turns columns into capped columns. Figure 1 shows ice
particles sampled in the same MCS in convective regions, but at different temperatures. Even though these images were sampled in horizontally separated regions, they support the proposed sequence of ice metamorphosis. Analysis of ice particle images HIWC regions in other studied MCSs showed the same altitude pattern of changes of particle habits.

A conceptual diagram describing the metamorphosis of ice particles in convective regions of
studies MSCs is shown in Fig. 2. It should be noted that HIWC regions frequently contain ice particles with habits different from columns or capped columns. Presence of such particles is related to the entrainment in the convective updrafts of aged ice particles with different histories from the ambient cloud regions or precipitating from above.

**2.4 Ice concentration, particle sizes and IWC**

Besides the shape of ice crystals, the way mass is distributed across the population of ice particles may also be exploited to retrieve the microphysical processes which the ice particles underwent. Thus, the relationship between $IWC$, $N_{ice}$ and particle size may provide important information about the mechanisms of HIWC formation. In the following discussion, the $MMD_{ice}$
will be used as a metric of the population of ice particles.

To explore $IWC - N_{ice}$ and $IWC - MMD_{ice}$ relationships, we employed data collected by particle imaging probes, primarily the 2DS and PIP. To reduce the biases in calculations of $N_{ice}$ due to uncertainty in the definition of the depth-of-field and diffraction effects on particle image sizing, the first three size bins in the 2DS measurements were skipped which resulted in
PSD and $N_{ice}$ calculations for $D_{ice} \geq 40$ μm. Such truncation is a reasonable compromise between uncertainty in concentration related to measurements of low pixel number images and underestimation of particle concentration due to disregarding small images.

Analysis of the mass distributions of ice particles shows that the particle mass concentration rapidly decreases with the decrease of particle size for $D < 200$ μm. The estimated bias in
calculations of $MMD_{ice}$ related to uncertainty of particle counting with $D < 40$ μm for the Cayenne data set is expected to not exceed 8%.

The $IWC$ was primarily measured by the IKP. Calculation of $IWC$ from the IKP required accurate measurements of the background humidity. Unfortunately, in several flights the background humidity sensor was malfunctioning. Therefore, to increase the statistics of $IWC -$
$N_{ice}$ and $IWC - MMD_{ice}$ relationship, $IWC$ was calculated from integration of the composite PSD measured by 2DS and PIP. The coefficients $a$ and $b$ in the size-to-mass parameterization



$M = aL^b$, where $M$ is the particle mass and $L$ is the particle image maximum dimension, were found by minimizing the difference between $IWC$ calculated from the measured PSD and that measured by the IKP for the flights when accurate $RH$ measurements were available.

Figure 3a shows probability density functions (PDF) of $N_{ice}$ for ice particles with $D_{ice} \geq 40$ μm in HIWC cloud regions with $IWC \geq$1 g m$^{-3}$, 1.5 g m$^{-3}$ and 2 g m$^{-3}$. As it is seen the PDFs of $N_{ice}$ have modal distributions with modal concentrations increasing approximately from 100 L$^{-1}$ to 250 L$^{-1}$ with increase of $IWC$. The ice concentration in HIWC cloud regions varies across a broad range reaching maximum of 4 × 10$^3$ L$^{-1}$.

The PDFs of $N_{ice}$ with $IWC <$ 1 g m$^{-3}$, 0.5 g m$^{-3}$ and 0.1 g m$^{-3}$ are shown in Fig. 3b. As it is seen the behaviours of $N_{ice}$ PDFs in the low IWC regions contrasting to those in the HIWC environments in Fig. 3a. The ice particle concentrations in low $IWC$ clouds appear to be much smaller compared to that in HIWC regions. Comparisons of 10$^{th}$, 50$^{th}$ and 90$^{th}$ percentiles of $N_{ice}$ for low IWC and HIWC environments in Table 1 illustrate the well pronounced difference

between these low and high IWC cases.

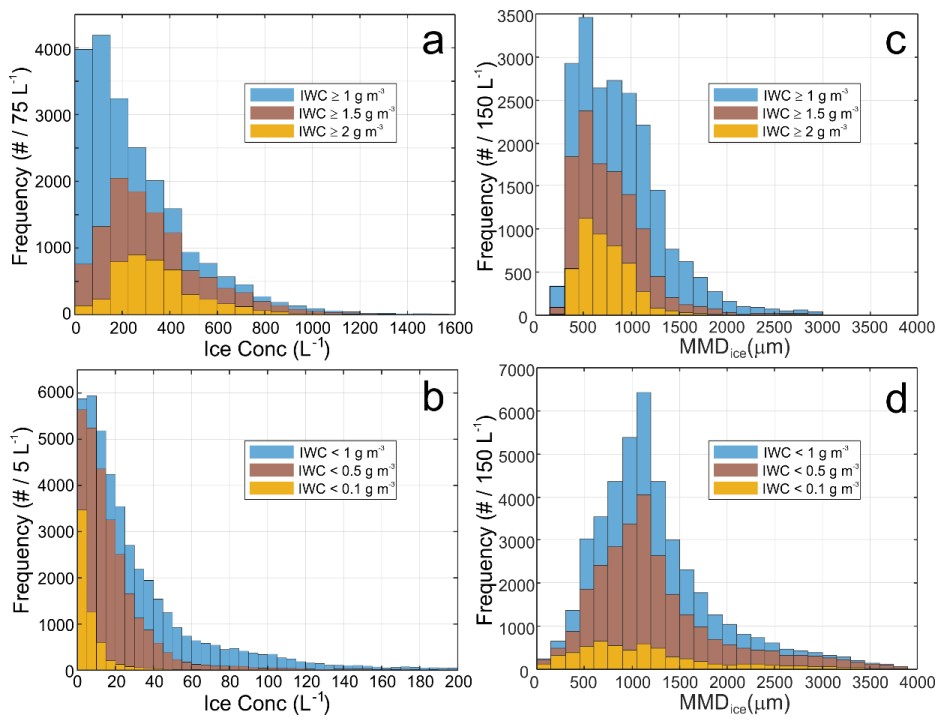

**Figure 3**. Probability density function of $N_{ice}$ and $MMD_{ice}$ in cloud regions with (a, c) $IWC \geq$1 g m$^{-3}$, 1.5 g m$^{-3}$ and

2 g m$^{-3}$ and (b,d) $IWC <$ 1 g m$^{-3}$, 0.5 g m$^{-3}$ and 0.1 g m$^{-3}$, respectively. Measurements were collected in the temperature range -15°C $< T <$ -5°C.

The PDFs of $MMD_{ice}$ in high and low IWC cloud regions are shown in Figs. 3c,d. As it is seen $MMD_{ice}$ in HIWC cloud regions cloud ice particles have smaller $MMD_{ice}$, whereas in low IWC

clouds the $MMD_{ice}$ is higher. The relationships between $MMD_{ice}$ and $IWC$ can also be seen from Table 1, which shows increase of MMDs' 10$^{th}$, 50$^{th}$ and 90$^{th}$ percentiles with the decrease of IWC.





**Table 1.** $10^{th}$, $50^{th}$ and $90^{th}$ percentiles of ice particle concentrations with $D_{ice} \geq 40$ μm and MMD in cloud regions with different IWC in the temperature range -15°C< $T$ <-5°C.

| IWC range | ice concentration (cm⁻³) | | | $MMD_{ice}$ (μm) | | |
|---|---|---|---|---|---|---|
| | $10^{th}$ percentile | $50^{th}$ percentile | $90^{th}$ percentile | $10^{th}$ percentile | $50^{th}$ percentile | $90^{th}$ percentile |
| IWC > 2 g m⁻³ | 162 | 319 | 600 | 436 | 687 | 1070 |
| IWC > 1.5 g m⁻³ | 103 | 285 | 626 | 403 | 711 | 1200 |
| IWC > 1 g m⁻³ | 44 | 203 | 580 | 407 | 819 | 1480 |
| IWC < 1 g m⁻³ | 3.8 | 21 | 92 | 562 | 1120 | 2180 |
| IWC < 0.5 g m⁻³ | 2.6 | 13 | 42 | 554 | 1100 | 2320 |
| IWC < 0.1 g m⁻³ | 0.9 | 4.2 | 18 | 359 | 1020 | 2420 |

The relationship between $N_{ice}$, $MMD_{ice}$ and $IWC$ can be seen in more details from the scatter diagrams of $N_{ice}(IWC)$ and $MMD_{ice}(IWC)$ in Figs. 4a,b. As shown in Fig. 4a, the median $N_{ice}$ increases monotonically with the increase of $IWC$ up to 2 g m⁻³ and then remains approximately constant, equal to 320 L⁻¹. However, $MMD_{ice}$ has a general tendency to decrease with increase of IWC (Fig. 4b).

The main conclusion from Figs. 3 and 4 is that despite of a large scattering of $N_{ice}$ and $MMD_{ice}$ vs $IWC$, HIWC environments at -15°C < $T$ < 5°C mainly consist of a relatively high concentrations of small ice particles with the median values of $N_{ice}$ ranging from 200 L⁻¹ to 300 L⁻¹ and median $MMD_{ice}$ ranging between approximately 700 μm and 800 μm (Table 1). However, in low IWC clouds most cases have a relatively low $N_{ice}$ with median values less than 20 L⁻¹ and median $MMD_{ice}$ exceeding 1000 μm. As was shown in Hu et al. (2021), the correlation between $N_{ice}$ and $IWC$ remains down to the -50°C < $T$ < -40°C temperature range.

There are two possible explanations for the observed relationship between $N_{ice}$ and $IWC$. First, ice particles could be initiated at the upper parts of the MCS through homogeneous (e.g., at $T$ <-40°C) or heterogeneous (at $T$ >-40°C) freezing of cloud droplets in convective updrafts. The amount of generated ice may be sufficient to explain the observed concentrations of ice particles. The drawback of this explanation is that the shape of ice particles formed on frozen droplets at low temperatures is dominated by bullet-rosettes (e.g., Bailey and Hallett, 2009). However, this type of ice particle habit was not observed in any noticeable amounts in the studied MCSs. This makes this explanation less plausible.

The second explanation is related to ice initiation through SIP processes above the melting layer where secondary ice particles are then transported to the upper parts of the MCS by convective updrafts. This explanation is consistent with the observed ice particle habits as discussed in section 2.3.





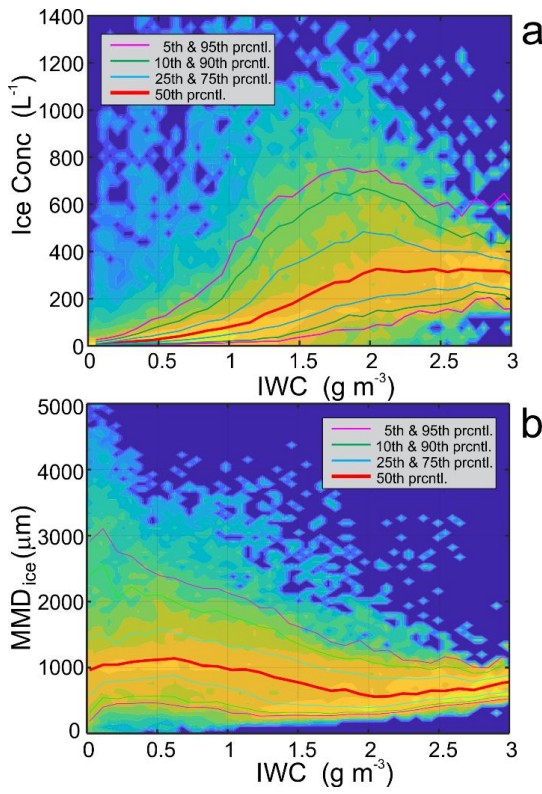

**Figure 4**. Scatter diagram(a) $N_{ice}$ vs $IWC$ and (b) $MMD_{ice}$ vs $IWC$ at -15°C<$T$<-5°C. In the density map the PDFs of $N_{ice}$ and $MMD_{ice}$ were normalized on unity in each interval $\Delta IWC$ = 0.05 g m$^{-3}$ in the temperature range -15°C< $T$ <-5°C.

### 2.5 Mixed-phase cloud regions

The analysis of the data collected from the NRC CV580 showed that in mature MCSs mixed-phase environments are quite sparse. The spatial occurrence of mixed-phase regions averaged over the entire data set at -2°C> $T$ >-15°C was 4.8%. Mixed-phase cloud was observed in spatially small-scale regions with the mean and median horizontal extension of 0.48 km and 0.23 km, respectively, and the maximum length of 2.5 km. Nearly all observed mixed-phase cloud regions were associated with vertical updrafts exceeding 1 m s$^{-1}$. However, many strong updrafts with $u_z$~ 10 m s$^{-1}$ contained no liquid. Average and median LWC observed in mixed-phase regions was 0.047 g m$^{-3}$ and 0.034 g m$^{-3}$, with maximum LWC peaking up to 0.1 g m$^{-3}$. For most of the observed mixed-phase cases, $LWC << IWC$ with an average liquid water fraction $LWC/TWC$= 0.13. These results are generally consistent with the data collected from the SAPHIRE F20 (Strapp et al., 2021). The F20 data showed 0.2% of spatial occurrence mixed-phase at -30°C and no presence of liquid below -35°C. Earlier a similar conclusion about trace amounts of liquid water in tropical MCSs was obtained in Lawson et al (1998).

The low occurrence and sparsity of mixed phase in MCSs is consistent with the observation of high concentration of ice in HIWC regions, which creates conditions favorable for rapid glaciation of clouds. Thus, for a still air $U_z$=0m s$^{-1}$, initial $LWC_0$=0.1 g m$^{-3}$, temperature $T$=-5°C, ice particle size $L_i$ = 100 µm and concentration $N_i$ = 300 L$^{-1}$ the mixed-phase cloud will glaciate





due to the Wegener–Bergeron–Findeisen process within 120 seconds (Korolev and Isaac, 2003). The riming process will expedite conversion of liquid into ice and the glaciation time will be even shorter.

Besides the concentration of ice, the glaciation time also depends on vertical velocity and the initial LWC (Korolev and Isaac, 2003; Pinsky et al., 2014). Since the droplets and ice may grow simultaneously (Korolev and Mazin, 2003), the glaciation process may take longer or the cloud may glaciate after reaching the homogeneous freezing temperature. Therefore, the mixed-phase cloud may exist only in a horizontally narrow regions associated with convective updrafts. Analysis of the X- and W-band radar Doppler velocities showed that the typical
horizontal dimensions of the updrafts are a few hundred meters. This is consistent with the typical spatial horizontal scale of mixed-phase clouds in MCSs.

### 2.6 Sparse supercooled large drops

One of the interesting features related to mixed-phase regions is the observation of SLDs
with $D$ ranging from 50 $\mu$m to 200 $\mu$m embedded in ice clouds with no presence of small droplets. An example of spatially sparce large drops is shown in Fig. 5 (red frames).

Identification of "sparse" SLD regions in tropical MCSs with the help of OAPs (e.g., 2DS, CIP, 2DC) and scattering probes (e.g., FSSP, CDP) is hindered due to the limitations of their performances. Thus, scattering probes do not respond in sparse SLD regions since, in most
cases, the diameters of the smallest drops there exceed the maximum measured size of these instruments. On the other hand, OAPs images of out-of-focus small ice crystals, in many cases, appear as the out-of-focus images of drops, and therefore images of ice particles may be confused with liquid drops.

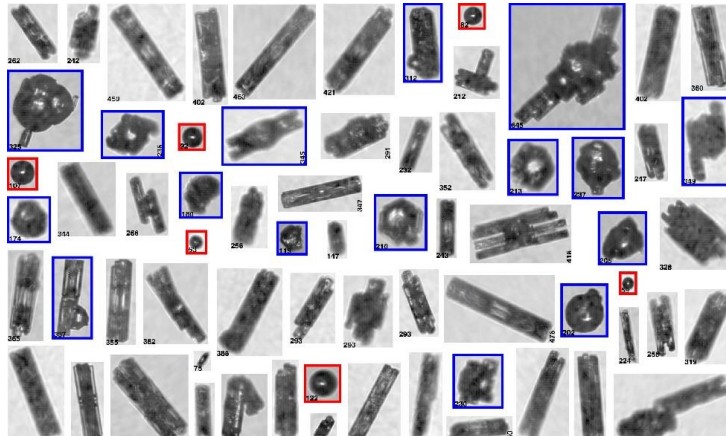


**Figure 5**. Spatial sequence of cloud particle images measured by the CPI with isolated large drops (red frames) embedded in the ice environment shown in Fig. 6. Blue frames indicate ice particles with attached or embedded large frozen drops. $H$=6500m, $T$=-9.5°C, 10:48:10-10:48:11 UTC, 26 May 2015.

In this study, the detection of sparse SLDs was identified from the CPI high-resolution imaging probe with the help of the image processing software trained on the ensemble of droplet images collected in warm clouds. Even though the CPI is not well qualified to measure bulk microphysical parameters, it allows for a crude estimate of the number concentration, which is sufficient for this study. Assessment of the number concentration based on Connoly et





al. (2007) showed that the droplet concentration in sparse SLD regions varied in a wide range
      with a maximum of the order $10^2$ L$^{-1}$. The estimate of the minimum local droplet concentration
      depends on the spatial averaging scale and it may go well below $10^{-3}$ L$^{-1}$.
        Figure 6d shows a time series of SLD concentration with drop diameter larger 60 μm, 80 μm,
      100 μm, and 200 μm. As can be seen sparse SLD regions were usually observed in convective
updrafts and their vicinity (Figs. 6b,c). In many cases in tropical MCSs, SLD regions (Fig. 6d) are
      spatially separated from cloud regions with small droplets measured by scattering probes (Fig.
      6e).
        Strictly speaking, the identification of the thermodynamic phase of spherical particles from
      their images at $T < 0$°C is ambiguous since the images of liquid and freshly frozen non-deformed
drops appear the same way. However, the response of RID in the sparse SLD regions (Fig. 6f),
      unambiguously indicate that at least some fraction of the spherical particles in the CPI imagery
      is in liquid phase. Based on the rate of changes of the RID signal (Mazin et al., 2001) it was
      found that in SLD regions at 2 km above the melting layer LWC may reach over 0.1 g m$^{-3}$. It is
      worth mentioning that there were multiple episodes of airframe icing experienced by the
CV580 in the SLD regions during operations in tropical MCS, two of which resulted in occasions
      of buffeting (personal communication NRC pilot A. Brown).
        A conventional explanation of large drops consists of attributing their formation to collision-
      coalescence process. Such explanation assumes the presence of small droplets to enable the
      growth of larger drops due to coalescence with smaller ones. The presence of small droplets
will also lead to the formation of rimed ice particles. Rimed ice can be easily identified from the
      high-resolution imagery based on the distinct features on ice particle surfaces. However, no
      small-droplet riming was found within hundreds of meters around regions with sparse SLDs.
      Instead, there were many large (50-200 μm) frozen drops on the surfaces of ice particles or
      regrown in ice crystals (blue frames in Fig. 5). This is suggestive that the collision-coalescence
process was unlikely to have a significant contribution to the formation of sparse SLDs.
        An alternative explanation of the sparse large drops is related to their uplift through the
      melting layer. This hypothesis is supported by the absence of small droplet riming and
      observation of large frozen drops inside or attached to ice particles (Fig. 5, blue frames). The
      terminal fall velocity of a 200 μm drop at the level of the melting layers is approximately 1 m s$^{-1}$.
Therefore, these drops could be transported by a moderate convective updraft (e.g., 4-5 m s$^{-1}$)
      from the melting layer to the level of the observations ($\Delta H \sim 1$ km) within 5-6 min.

   **2.8 Disturbance of the bright band**
        Another interesting feature observed during the HAIC-HIWC field campaign is a
disturbance of the top of the bright band. Figure 6a shows one of the examples of the bright
      band patterns measured by the X-band radar during transit through the MCS on 26 May
      2015 (Appendix A). As seen in Fig. 6a, the top of the bright band was relatively uniform in
      stratiform regions where at the flight level the vertical wind velocity $U_z$ is close to 0 m s$^{-1}$
      (Fig. 6c) (time segments: 10:44:40-10:46:30; 10:52:10-10:53:10; 10:58:20-11:01:00).
520   However, in convective regions the enhanced reflectivity above the level of undisturbed
      bright band may extend up to 1.5-2 km. This can be seen from the diagrams with the
      vertical Doppler velocity (Fig. 6b) and vertical wind velocity $U_z$, at the flight level (Fig. 6c),
      which correspond to the convective regions within the time segments 10:46:4510:48:40;
      10:49:50-10:51:30; 10:53:20-10:53:25.



525

**Figure 6**. Time series of (a) X-band radar reflectivity, (b) Doppler velocity measured by X-band radar, (c) AIMMS-20 and RSM-858 vertical wind velocity, (d) SLD number concentration estimated from CPI measurements, (e) FSSP and CDP concentration, (f) Rosemount Icing Detector frequency. The measurements were collected at 6500 m <*H*< 7000 m, -12°C < *T*< -9.5°C, 26 May 2015.

530



The bright band forms as a result of enhanced radar return from melting ice particles falling below the freezing level. The linkage between the bright band and freezing level suggests that the spatial vertical fluctuations of the bright band may be explained by the vertical fluctuations of the 0°C-isotherm. However, 1.5-2 km vertical changes of the altitude of the freezing level would result in spatial fluctuations of the temperature, $\Delta T$, of the order of 9°C to 12°C, with the assumption of a moist adiabatic lapse rate. Such temperature fluctuations are overly high and were never observed on kilometer spatial scales. Typically, during leveled flights in convective regions in studied MCSs, the amplitude of temperature fluctuations did not exceed ±1°C.

A potential explanation of the disturbance of the bright band in convective regions is related to recirculation of completely and partially melted ice through the melting layer. The precipitation size drops formed after the melting of ice particles may be transported above the melting layer with updrafts generated by regular convection or gravity waves. Above the melting layer SLDs will interact with the preexisting ice, and thus produce high density spherical particles and graupel, which together with SLDs will result in the enhancement of reflectivity.

The above explanation of the enhanced reflectivity over the melting layer is consistent with the observation of SLDs (section 2.6). It is also supported by positive Doppler velocity observed below and withing the melting layer as in Fig. 6b (10:49:11-10:49:40, 10:52:50-10:53:20; 10:56:02-10:56:12). It is worth mentioning that the radar average Doppler velocity has a higher sensitivity to larger drops. Therefore, even though the average Doppler velocity is negative, small droplets driven by updrafts may have positive vertical velocity.

**2.9 High IWC cloud regions and vertical updrafts**

Observations collected during the HAIC-HIWC campaign showed that traversing through convective updrafts was usually accompanied by an increase of $IWC$ over 1 g m$^{-3}$. Nearly all convective regions with $u_z$ >2 m s$^{-1}$ observed during the Cayenne campaign were associated with enhanced IWC. The HIWC regions around convective updrafts extended from a few hundreds of meters to tens of kilometers. A detailed study of the statistics of horizontal spatial scales of HIWC cloud regions is available from Strapp et al. (2021). However, in some cases no convective updrafts were observed during transit through the HIWC regions.

The time series in Fig. 7 illustrates relationship between $IWC$ and $u_z$ described above (Appendix A). As it is seen, four of five HIWC cells (i.e., #2-5 in Fig. 7b) are associated with the embedded convection observed somewhere within these cells (Fig. 7c). In-situ vertical velocity measured on the flight level (Fig. 7c) correlates well with the Doppler velocity (Fig. 7d), which in some cases may extend to the cloud top of the MCS. However, no distinct convective updrafts were observed within HIWC cell #1 (in Fig. 7b; ~10:20-10:26). An encounter of the convective updrafts during the transit through the HIWC region depends on the flight pattern and it is possible that the aircraft trajectory did not intersect or came close enough to the convective region. It is also possible that during sampling of the HIWC region the convection in this area faded away.



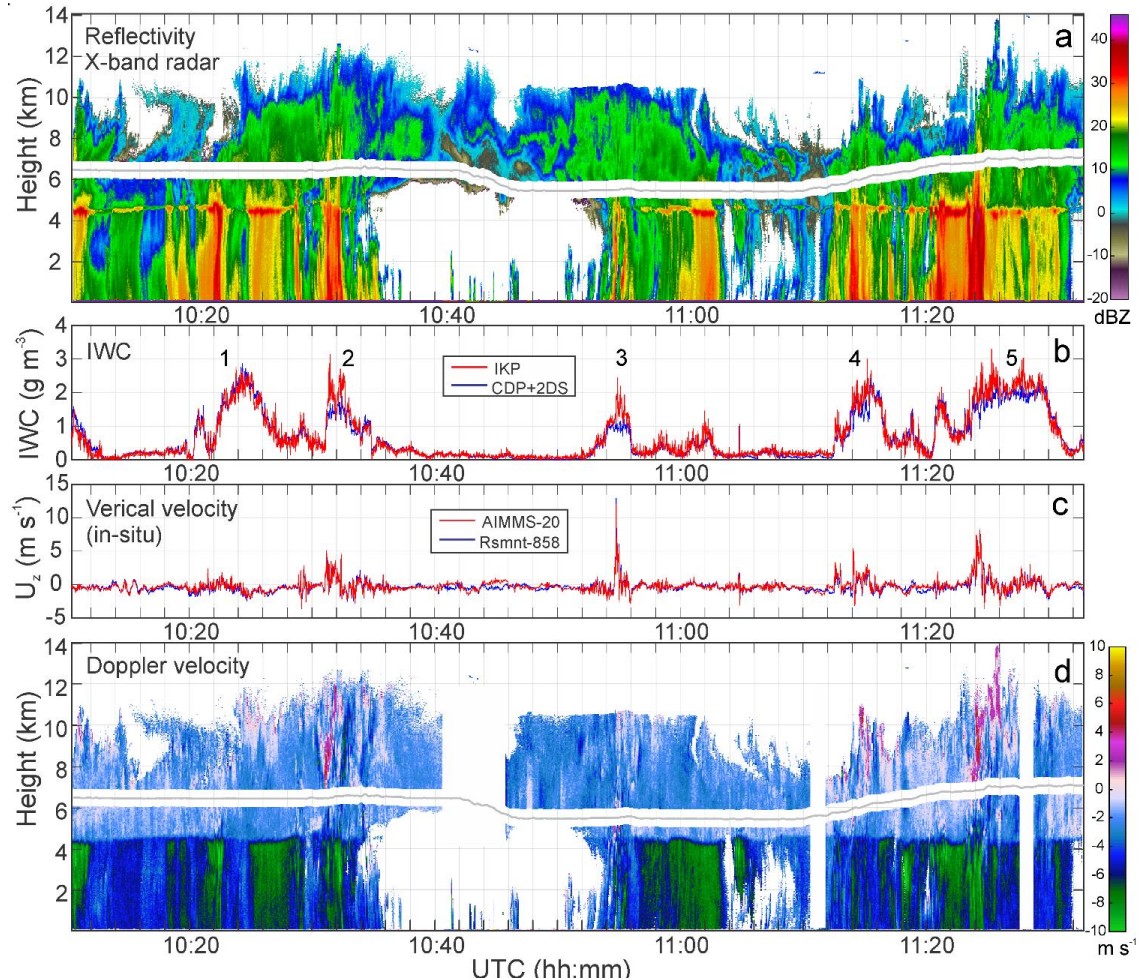

**Figure 7**. Time series of (a) X-band radar reflectivity, (b) IWC, (c) vertical velocity, (d) Doppler velocity measured by the X-band radar. The numbers in (b) indicate HIWC cell number (see text). The measurements were collected from the NRC Convair-580 on 27 May 2015.

A comprehensive analysis of the relation between $IWC$ and proximity to the center of the nearest convection based on the combined analysis of the in-situ and satellite data was conducted by Hu et al. (2021). It was found that $IWC$, on average, depends on the distance from the convective core ($L_{conv}$), and that $IWC$ decreases with increasing $L_{conv}$. The dependence of IWC on the distance from the convective core was also indicated in Lawson et al. (1998).

The relationship between $IWC$ and $U_z$ described above suggests that the convective updrafts transport small ice particles to the upper levels. Due to the small fall velocity newly formed secondary ice particles can be transported to high altitudes even with a relatively low vertical velocity. During the ascent, ice crystals grow through water vapor deposition which ultimately results in the formation of HIWC.



### 3.    Numerical simulations

The objective of this section is to summarize results of numerical simulations of HIWC in tropical MCSs and explore the role of SIP in the vicinity of the melting layer on the formation of HIWC environments.

### 3.1  Model configuration

#### 3.1.1   Atmospheric model

The numerical model used in this study is the Global Environmental Multiscale (GEM) atmospheric model (Côté et al., 1998; Girard et al., 2014).  GEM is the operational NWP model of Environment and Climate Change Canada (ECCC), used for all of ECCC's deterministic and ensemble NWP prediction systems. It is also capable at running at high spatial resolution (including sub-km, near cloud-resolving scales) for detailed atmospheric research purposes. The dynamical core of GEM is formulated based on the non-hydrostatic, fully-compressible, primitive equations with a terrain-following hybrid vertical grid. For the simulations described below, the model dynamics and physics options are summarized in Appendix B (Table B1).

A tropical MCS that was observed during the HIWC-HAIC campaign was simulated using a quasi-idealized model configuration of GEM, similar to the simulations described in Qu et al. (2022). An analysis sounding from ECCC's global deterministic NWP system was used to initialize GEM on a domain with a 250-m horizontal grid spacing, with 256 x 256 km$^2$ and 183 unevenly-spaced vertical levels. The initial atmospheric conditions were horizontally homogeneous, based on the sounding taken 1200 UTC 15 May 2015 at 6.769°N and 49.551°W (Appendix B). The initial sea surface temperature was set to 300.37 K. To initiate the convection, the updraft nudging method of Naylor and Gilmore (2012) was used to force convection during the first 15 min of the simulation with maximum vertical wind speed of 10 m s$^{-1}$. Three distinct updrafts 16 km apart from each other in the western part of the simulation domain were initialized. The simulations were integrated for 3 h.

In order to examine the impact of SIP on HIWC from a modeling point of view, two simulations were run. In the baseline (i.e. control) simulation, referred to BASE, four ice categories in the microphysics scheme (see 3.1.2 below) were used and all SIP processes were deactivated. In the second run, referred to as SIP-ON, the parameterized HM and FFD mechanisms for SIP were activated (see 3.1.3 below). A comparison of the two simulations allows for an investigation of the impacts of SIP processes on HIWC.

Note that at 250 m, the simulations conducted are near to what is conventionally regarded as "cloud resolving" or large eddy simulation (LES) scale (with horizontal grid-spacings of ~100 m or less). At this scale, a 3D turbulence parameterization should be used. GEM currently lacks the capability to run in true LES mode, thus the following results are not truly cloud-resolving simulations.  However, GEM has been used successfully by other groups to simulate deep convection with a 250-m horizontal grid spacing (e.g. Belair et al., 2018).  Furthermore, the model results shown below are reasonably close to the in-situ observations in terms of bulk microphysical fields and the model sensitivity with respect to the inclusion of SIP processes is consistent with observational evidence. Hence, despite the limitations of the model, the simulation results can be used with a sufficient degree of confidence to support the conceptual model proposed in section 5.

#### 3.1.2  Microphysics scheme

Given the relative importance of the model microphysics in this study, some additional detail on the P3 microphysics scheme is warranted. The P3 scheme was introduced in Morrison and Milbrandt (2015). Since its inception there have been several major developments, summarized



in Cholette et al. (2023). The liquid-phase component of P3 uses a standard two-moment, two-category approach, with liquid hydrometeors partitioned into "cloud" (small droplets) and "rain". Each liquid category is represented by a complete gamma size distribution and has prognostic variables for the mass and number mixing ratios. As such the size distribution of each category goes from zero to infinity; however, for practical purposes a diameter of around 80 microns can be regarded as a de facto demarcation size between the two categories.

The ice phase, on the other hand, is treated much differently in P3 than in traditional schemes which typically partition frozen hydrometeors into several predefined categories (e.g. pristine "ice", "snow", "graupel", etc.) with associated fixed parameters to prescribe physical properties (e.g. density). Rather, P3 has a user-specified number of generic "ice" categories, each of whose physical properties can evolve freely and continuously in time and space due to the choice of prognostic variables (Milbrandt and Morrison, 2016). The PSD for each ice category is a complete gamma function. The following six mixing ratio variables for each ice category are predicted: the total mass, rime mass, liquid mass, total number, rime volume, and the sixth moment. With these prognostic variables, which are advected by the GEM dynamics and whose values are updated due to microphysical process in the P3 scheme itself, bulk physical properties (of each ice category) including density, mean size, rime fraction, liquid fraction (thereby allowing for mixed-phase particles), PSD spectral width, etc. are all predicted independently. Fields such as bulk fall speeds, radar reflectivity, etc. can also be computed and benefit from the degrees of freedom.

While each "ice" category is generic and can represent a wide range of mixed-phase hydrometeor particles, it can only represent a single dominant mode or particle type (e.g. lightly-rimed small aggregates) for the population of particles that it represents. In order to represent multiple modes of ice particles at a given point in space (i.e. model grid point) and time, two or more ice categories must be used. Thus, in order to model SIP using P3 at least two ice categories must be used since SIP can result in the production of numerous small crystals in the presence of pre-existing larger ice, such as graupel in the case of the HM SIP process. The P3 microphysics scheme, therefore, is an appropriate and useful tool for the examination of the impacts of SIP in an atmospheric model. Qu et al. (2022) showed a near "convergence" of GEM model solutions with P3 using three ice categories. In the simulations in this study, four ice categories were used.

### 3.1.3 Parameterization of SIP processes

Two SIP mechanisms were examined, HM and FFD, and are activated in the SIP-ON simulation. The parameterized HM process produces 10 μm ice splinters with maximum number of 350 crystals per mg of collected liquid cloud droplets during riming within a temperature range of -3°C > $T$ > -8°C, with the peak value at -5°C and varying linearly to 0 at the extreme temperature ranges.

For FFD, the number of ice splinters produced is calculated based on the number of raindrops with diameters between 100 and 3500 μm colliding with ice particles with diameter smaller than 100 μm. This parameterized process follows the relationship proposed by Lawson et al. (2015), based on agreement of parcel model results with in-situ aircraft observations:

$$N_{sip} = N_r 2.5 \times 10^{-11} D^4 \tag{1}$$

where $N_{sip}$ is the number of ice splinters produced by the freezing of rain drops of a given diameter $D$. $N_r$ is the number of drops with diameter $D$ colliding with ice particles with





diameter smaller than 100 μm. The total number of ice splinter produced by FFD process will be the integral of Eq. 1 over $D$ between 100 and 3500 μm.

Based on Keinert et al. (2020), the FFD process exhibit activity within a specific temperature
range of -25°C < $T$ < -2°C. Their findings indicate that among supercooled liquid drops, the most frequent splitting and ejection events occurr at -12.5°C. However, the dependence of the total number of small ice splinters versus temperature and drop size remains unclear from laboratory studies to date. The observational evidence suggests that SIP production is particularly active within 2 km above the melting layer. In this study, the maximum FFD rate
$N_{sip}$ is set to occur at $T$ = -4°C. Beyond this point, the FFD rate is assumed to decrease linearly from its peak at -4°C to zero at the two temperature extremes (-25°C and -2°C).

In the analysis in the following sections, various quantities are computed from direct model output fields from the GEM simulations. The raw model output fields include mixing ratios for hydrometeor mass and total number for cloud droplets ($Q_c$ and $N_c$), rain ($Q_r$ and $N_r$), and each
category $x$ for ice ($Q_{i,x}$ and $N_{i,x}$). For $Q_{i,x}$, this is the total (for category $x$) of the deposition mass, rime mass, and liquid (for mixed-phase particles) mass, each of which are independent prognostic model variables. With the air density $\rho_a$, the mass contents are computed as $LWC_{cloud} = \rho_a Q_c$, $LWC_{rain} = \rho_a Q_r$, and $IWC = \rho_a(Q_{i,1} + Q_{i,2} + Q_{i,3} + Q_{i,4})$ and the total number concentrations as $N_{cloud} = \rho_a N_c$, $N_{rain} = \rho_a N_r$, and $N_{ice} = \rho_a(N_{i,1} + N_{i,2} + N_{i,3} +$
$N_{i,4})$. However, for consistency with the observations, which exclude ice particle sizes smaller than 40 μm, each $N_{i,x}$ was re-computed from the ice PSD as the incomplete integral from 40

μm to infinity. The mean-mass particle sizes were computed as $D_{3\,cloud} = \left(\frac{6}{\pi \rho_w} \frac{LWC_{cloud}}{N_{cloud}}\right)^{1/3}$,

$D_{3\,rain} = \left(\frac{6}{\pi \rho_w} \frac{LWC_{rain}}{N_{rain}}\right)^{1/3}$, and $D_{3\,ice} = \left(\frac{6}{\pi \rho_{ice}} \frac{IWC}{N_{ice}}\right)^{1/3}$, where $\rho_w$ and $\rho_{ice}$ are the bulk densities of water and ice, respectively. Note that $D_{3\,ice}$ represents the diameter of an equivalent-mass
sphere. In P3, $\rho_{ice}$ is a predicted variable (for each category), however since we examine primarily small, unrimed ice a value of 917 kg m⁻³ is assumed, for simplicity. The model reflectivity, $Z$, is computed in-line (within the microphysics scheme); see Cholette et al. (2023) for details. The total SIP rates, ($dN_{ice}/dt$), are computed in-line and output for analysis.

**3.2  Comparisons of SIP-ON and BASE simulations**
The two sets of vertical cross-sections of SIP-ON and BASE in Fig. 8 show stark differences between the model fields of the SIP-ON and BASE simulations. To facilitate comparison, both SIP-ON and BASE cross-sections were produced along the same vertical planes at 90 min simulation time. The top view of the cross sections' location within the cloud domain is shown
in Fig.C1 (Appendix C).

In the BASE simulation $N_{ice}$ increases by approximately two orders of magnitude within a short distance above an altitude of 10 km (Fig.8b). A similar rapid vertical changes of ice particle sizes is seen above $H{\sim}10$ km, where $D_{3\,ice}$ decreases from approximately 1-1.5mm at $H < 10$ km to 100-400 μm at $H > 10$ km (Fig.8d).





**Figure 8**. Vertical cross section domains of different parameters of GEM simulations for SIP-ON (left column) and BASE (right column) setups: (a,b) $N_{ice}$; (c,d) $D_{3\,ice}$; (e,f) $LWC_{cloud}$; (g,h) $IWC$; (i,j) $Z$; (k,l) $U_{Doppl}$. The cross sections were performed for the same vertical plane at 90 minutes on the modelled time. Horizontal dashed lines show levels corresponding to $T$ =0°C ($H$ =4.9km) and $T$=-40°C ($H$=11.1km) levels.



The vertical behaviour of $N_{ice}$ and $D_{3\,ice}$ in BASE is in contrast with those in the SIP-ON simulation. As seen from Fig. 8a,c, $N_{ice}$ and $D_{3\,ice}$ experience only modest changes in the vertical direction and they change in a smaller range compared to the BASE run. In the SIP-ON simulation, ice with large concentrations (~$10^3$ L$^{-1}$) is present in the cloud domain right above the melting layer (~5 km) and the field of $N_{ice}$ remains relatively uniform up to the cloud top (Fig. 8a).

Ice particle size $D_{3\,ice}$, on average, exhibits relatively minor changes between the melting layer and the cloud top (Fig. 8c). The $LWC_{cloud}$ field has broader horizontal extent (at least in most locations) and extends higher in BASE compared to the SIP-ON simulation (Figs. 8e,f). Figures 8g,h shows that the SIP-ON simulation has much higher values of IWC than BASE. Maximum IWC in the SIP-ON run approaches 4 g m$^{-3}$ in some regions (Figs. 8g), whereas in BASE

the maximum IWC barely reaches 2.5 g m$^{-3}$ (Fig. 8h). The area with high IWC (> 1 g m$^{-3}$) in SIP-ON (Fig. 8g) has considerably more horizontal and vertical extent compared to BASE (Fig. 8h). Figures 8i and j show that the radar reflectivity is noticeably lower in the SIP-ON simulation than in the BASE simulation. Such behaviour of $Z$ reversed to that of $IWC$. The domains of the Doppler velocity (Figs. 8k,l) show that SIP-ON clouds are more dynamically active than those in

BASE. The pattern of the Doppler velocity field in Fig. 8k also indicates significant transport of condensed water from the melting layer to the cloud top in the SIP-ON simulation. The intense dynamics and vertical transport of ice will stimulate vertical mixing and homogenizing fields of $N_{ice}$ and $IWC$ in the SIP-ON simulation (Figs. 8a).

       While Fig. 8 provides a qualitative comparison of the results of the SIP-ON and BASE runs, a

more definitive comparison of the two simulations can be examined through temporal and spatial averaging of the modeling results, as shown in Fig. 9. The temporal averaging was done over 1 h, from 75 to 135 min of the model cloud lifetime, which include 13 modeling domains separated by 5 min. The spatial averaging was computed over the area of precipitation. The selection of the averaging domain above the melting layer was aimed at excluding the outflow

anvil area, which may extend to large distances from the convective region and is significantly affected by entrainment and mixing. Another reason for the choice of averaging was to mitigate biases when comparing with in-situ airborne observations, which were collected mainly in the vicinity of the convective parts of the MCSs.

       Figure 9a shows that below 11 km the mean IWC in the SIP-ON simulation is up to two times

higher than that of BASE. However, at $H$ > 11 km the average IWC in the SIP-ON and BASE simulations are approximately equal. The 99$^{th}$ percentile $IWC_{99}$ in SIP-ON remains systematically higher from the melting layer up to the cloud top. Maximum of the mean IWC occurs between 8 and 9 km. LWC in BASE remains consistently higher than that of SIP-ON between the melting and homogeneous freezing levels (Fig. 9b).

As seen from Fig 9c, the behaviour of $N_{ice}$ is very different between the two simulations. In the SIP-ON run, the mean $N_{ice}$ increases up to ~500 L$^{-1}$ in the first kilometer above the freezing level and then remains moderately constant, varying between 500 L$^{-1}$ and 700 L$^{-1}$ up to the homogeneous freezing level (~11 km). After the minimum at 11.5 km, $N_{ice}$ reaches maximum of ~1000 L$^{-1}$ above 14 km.





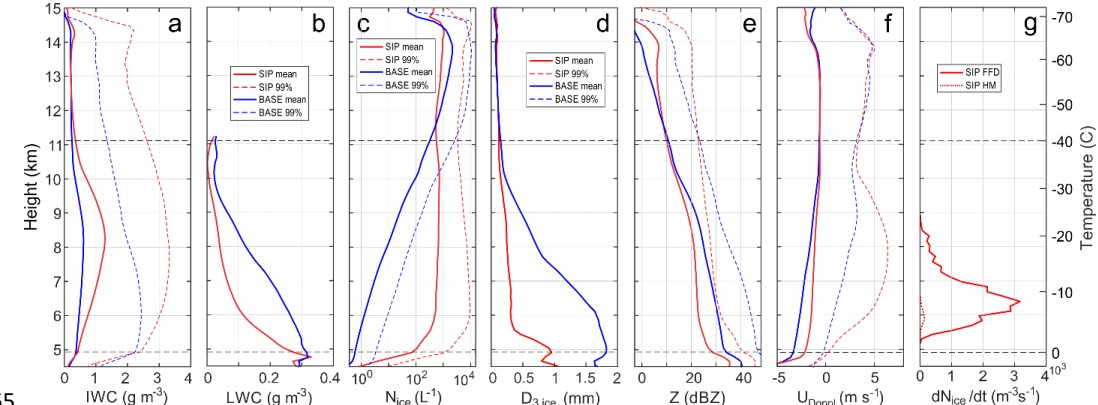

**Figure 9**. Average profiles of the results of SIP-ON (red) and BASE (blue) simulations. (a) $IWC$ (solid) and $IWC_{99}$ (dashed), (b) $LWC$ (solid), (c) $N_{ice}$ (solid) and $N_{ice\ 99}$ (dashed), (d) $D_{3\ ice}$ (e) $Z$ and $Z_{99}$; (f) $U_{Doppl}$ (solid) and $U_{Doppl\ 99}$; (g) ice crystal production rates for FFD (solid) and HM (dashed) for the SIP-ON simulation. The profiles spatially averaged over the cloud domain above the precipitation below the melting layer. The cloud domain was determined as the environment with $IWC > 0.001$ g m⁻³. The temporal averaging was performed over 13 modeling domains with time step 5 min within the time frame from 75 to 135 min.

The vertical distribution of $N_{ice}$ in BASE contrasts with the SIP-ON simulation. In the BASE run, $N_{ice}$ increases monotonically with altitude following a nearly exponential law and its values change from ~0.5 L⁻¹ at the freezing level to ~2200 L⁻¹ at an elevation of approximately 14 km. The rapid increase of $N_{ice}$ from 100 L⁻¹ to 2000 L⁻¹ between 10 km and 13.7 km occurs due to homogeneous freezing of cloud droplets in convective updrafts. As seen from Fig. 9b in both SIP-ON and BASE, convective updrafts are capable of maintaining liquid water up to the homogeneous freezing level. This results in an enhancement of $N_{ice}$ through the freezing of cloud droplets. The fact that the BASE $N_{ice}$ becomes larger than in SIP-ON above 11.5 km is explained by the larger number of cloud droplets transported by convection to the homogeneous freezing level in BASE compared to SIP-ON, which turn into ice and enhance ice concentration rather than evaporate due to the Wegener-Bergeron-Findeisen (WBF) process. This can also be seen in Fig. 9b, which shows a larger LWC in BASE above 11 km compared to the SIP-ON.

Figure 9d shows profiles of mean mass diameter $D_{3\ ice}$. As seen in both BASE and SIP-ON, $D_{3\ ice}$ decreases monotonically with increasing height up to 11 km, and $D_{3\ ice}$ in BASE always remains larger than in SIP-ON. However, above 11 km $D_{3\ ice}$ in both simulations is approximately equal and remains nearly constant up to the cloud top.

Comparisons of the radar reflectivity in Fig. 9e show that in both BASE and SIP-ON $Z$ decreases monotonically with altitude from the freezing level to the cloud top. However, the BASE $Z$ remains larger up to 11.5 km and then it becomes lower before reaching the cloud top. The difference in $Z$ between BASE and SIP-ON in the lower part of the cloud does not exceed 10 dBZ.

Figure 9f shows vertical changes of the Doppler velocity $U_{Doppl}$. The mean $U_{Doppl}$ in BASE remains smaller than in SIP-ON up to ~11 km and then both become approximately equal. The 99th percentiles $U_{Doppl}$ follow the same pattern as mean $U_{Doppl}$, i.e. BASE 99th percentile $U_{Doppl}$ is lower than SIP-ON 99th percentile $U_{Doppl}$ in the lower part of the cloud and they become equal above 11.5 km. Analysis of the vertical wind velocity $U_{air}$ shows that mean and 99th percentiles of BASE and SIP-ON $U_{air}$ remain approximately equal thought the cloud depth (Fig. C2 in



Appendix C). This indicates the dynamical similarity of the BASE and SIP-ON simulated clouds. Therefore, the lower values of the $U_{Doppl}$ in BASE can be mainly attributed to the presence of large particles and therefore, larger fall speeds compared to those in SIP-ON. This conclusion is also supported by larger BASE $D_{3\,ice}$ compared to SIP-ON $D_{3\,ice}$ (Fig. 9d).

Figure 9g shows the production rates of the FFD and HM processes in the SIP-ON simulation. Note that these processes were shut off in the BASE simulation. In the SIP-ON run, the FFD rate is much higher than that of HM and is therefore the main contributor to the production of secondary ice. The peak of the FFD rate occurs at approximately 6.4 km (-8°C).

It is worth noting that both SIP-ON and BASE simulations show distinct differences in the behavior of cloud microphysical parameters below and above the homogeneous freezing level (~11 km). Thus $IWC$, $D_{3\,ice}$, and $U_{Doppl}$ in both BASE and SIP-ON are approximately equal above 11 km. In contrast, below 11 km the changes of $IWC$, $N_{ice}$ $D_{3\,ice}$, $Z$ and $U_{Doppl}$ with altitude are vividly different between the two runs. This suggests a significant difference in dominant physical processes responsible for the formation of the cloud microstructure.

### 3.3 Comparisons with observations

In this section the results of the SIP-ON simulation are compared with the measurements collected from the NRC CV580 and SAFIRE F20 during the HAIC-HIWC Cayenne field campaign.

The outflow non-precipitating cloud segments and anvil regions identified from the onboard X-band radar were excluded from the CV580 statistics. The mean $LWC_m$, mean $IWC_m$, 99th percentiles $IWC_{99}$, and $D_{3\,ice}$ from the CV580 data, presented below, were sampled within the altitude range of 6 to 7.3 km and averaged over a total horizontal distance of 5374 km. The F20 $IWC_m$, $IWC_{99}$ and median mass diameter $MMD_{ice}$ for various altitudes were brought from Strapp et al. (2018, 2021).

The CV580 data were averaged over 1 s, corresponding to approximately 115 m spatial averaging at the average CV580 speed. Whereas the F20 data were averages over 0.5 nm (~930 m). The averaging scales of CV580, and F20 are close enough to that of the GEM model (250 m) to address the purposes of the comparisons. The differences in spatial averaging may affect the 99th percentile values. However, the mean values are expected to experience a minor effect.

For the purposes of comparisons, the results of the GEM simulations were averaged over a 1-h period from 75 to 135 min, as in Fig. 9. This time frame covers the "mature" period of simulated MCS after the maximum IWC has been reached, at approximately 45-50 min. During the averaged period, the cloud remains dynamically active with maximum updrafts peaking at $U_z \approx 15$m s$^{-1}$, although the convective area is diminishing over time. The choice of the averaging time of the simulation was aimed to align with the in-situ airborne CV580 and F20 sampling, which was typically extended over 2-3 h.

Figures 10a shows comparisons of the observed and modeled $IWC_m$ and $IWC_{99}$. The CV580 IWC were measured by the IKP and calculated from composite PSDs measured by 2DS and PIP. The F20 IWC was measured by the IKP (Strapp et al., 2020, 2021). The difference between measured and simulated $IWC_m$ and $IWC_{99}$ does not exceed 6% and 20%, respectively.

Figure 10b shows comparisons of the simulated $LWC_m$ to that calculated from the FSSP and CDP measurements. The difference between the modeled $LWC_m$ and that measured by the FSSP and CDP is 38% and 9%, respectively.





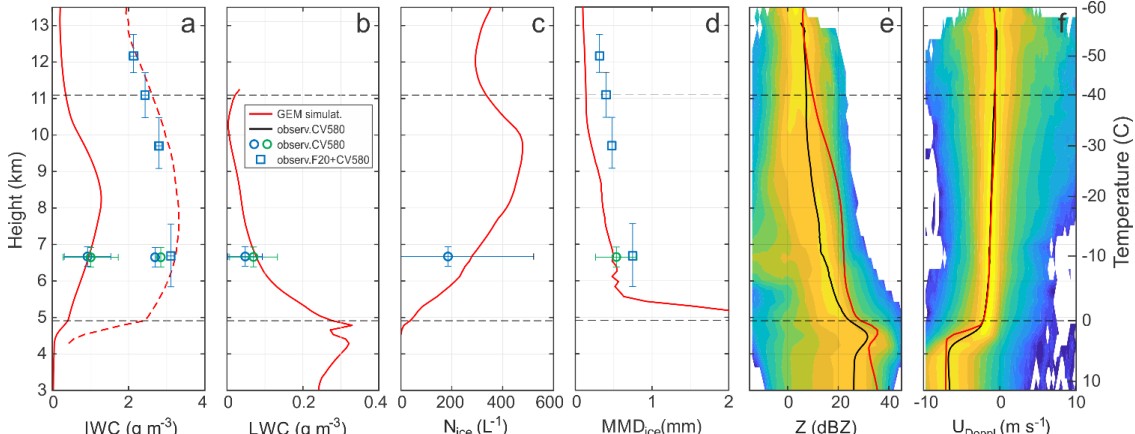

**Figure 10**. Comparisons of the average microphysical parameters from the GEM simulation and in-situ measurements collected from CV580 (circles) and F20 (squares). (a) simulated $IWC_m$(red, solid) and $IWC_{99}$, (red, dashed), measured by IKP (blue circles ) and 2D probes (green circle) (b) mean modelled $LWC$ (red), measured by FSSP (green circle) and CDP (blue circle) (c) mean modeled (red) and measured (blue circle) concentration of ice $N_{ice}$ with $D>40\mu m$, (d) modeled $D_{3\ ice}$ corrected on the ice density, CV580 $D_{3\ ice}$ (blue circles), measured F20 $MMD_{ice}$ (blue squares), (e) measured by the CV580 X-band radar probability density field and mean reflectivity $Z$ (black), and modeled mean $Z$ (red); (f) measured by the CV580 X-band radar probability density field and mean Doppler velocity $U_{Doppl}$ (black) and modeled mean $U_{Doppl}$ (red). The F20 vertical error bars indicate the temperature range of the sampled data. The CV580 vertical and horizontal error bars indicate the standard deviation of the sampling statistics.

For the comparison purposes, the model $N_{ice}$ was modified to match the size range of ice particles measured by the 2DS. Since the first three size bins in 2DS measurements were omitted due to the uncertainty of the depth-of-field definition and large errors in sizing of small particles, the 2DS measured $N_{ice}$ did not include ice particles smaller than 40 μm. In order to make direct comparisons of observed and modeled $N_{ice}$, the ice particle sizes smaller than 40 μm were also excluded from the modelled $N_{ice}$ by recomputing the concentration from the model ice PSD but with an incomplete gamma function, starting at 40 μm. Figure 10c shows a reasonable agreement between of the corrected modelled and measured $N_{ice}$, and the deviation between the modeled and observed $N_{ice}$ is 35%.

As mentioned in section 3.1, the modeled $D_{3\ ice}$ is the diameter of an equivalent-mass sphere. However, the measured ice particle sizes represent one dimension of randomly oriented projections of non-spherical particles. In order to make direct comparisons between the modeled and measured ice particle sizes, the simulated $D_{3\ ice}$ was modified from the mass-size relation

for unrimed ice in the original P3 microphysics scheme to $D_{3\ corr} = \left(\frac{\pi \rho_{ice}}{a} D_{3\ ice}^3\right)^{\frac{1}{b}}$, where $a$ and $b$ are parameters from Brown and Francis (1995). The comparisons of values measured from the CV580 and the corrected simulated $D_{3\ ice}$ is shown in Fig. 10d.

Figure 10d also shows median mass size $MMD_{ice}$ measured from F20. $MMD_{ice}$ and $D_{3\ ice}$ may be quite different depending on the shape of a PSD, which makes the comparisons of these two sizes not strictly valid. As seen from Fig. 10d, the modeled $D_{3\ ice}$ and measured $MMD_{ice}$ differs from each other from 2 to 4 times above 9 km. However, despite these differences in definitions of the modeled and measured sizes, the observed $MMD_{ice}$ profiles show a general trend in size reduction towards the cloud top, which is similar to the behaviour of simulated $D_{3\ ice}$ profile.





Comparisons in Fig. 10e show that the modeled mean $Z$ is systematically larger than the measured values, with differences reaching 10 dBZ. This is an overly large difference. The analysis of the model setup suggests that the complete gamma function for the ice PSDs in the microphysics scheme overestimates the number of large particle sizes, for which higher moments such as radar reflectivity are sensitive. At the time of writing, a solution for this diagnostic problem is being examined and the problem will be corrected in a future version of the P3 scheme. Note that small errors in the tail of the PSD can create relatively large errors in higher moments of the PSD (including $Z$) but only small errors in lower moments, such as those associated with the fields of interest (e.g. $N_{ice}$, $MMD_{ice}$, $IWC$, etc.), which appears to be the case in the simulations presented. Also, note that the model reflectivity field is computed as a simple diagnostic, not an instrument simulator per se; it is calculated based on Rayleigh scattering assumptions and does not account for Mie scattering for large particles for the case of X-band radar.

Despite the significant difference in reflectivity, the modeled and observed Doppler velocity values have a surprisingly good agreement. One of the explanations for such a good agreement is the weak dependence of particle fall velocity on particle size for diffusion grown ice particles (Pruppacher and Klett, 1997). In this case the presence of artificially large particles may not have a significant effect on the Doppler velocity.

Another set of comparisons is shown in Fig. 11. Figure 11a shows modeled and observed ice number concentration versus IWC. The latter was shown earlier in Fig. 4a. For comparison purposes, the particles < 40 $\mu$m were excluded from $N_{ice}$ to be consistent with the measured values. As seen in Fig. 11, the median observed and modeled ice concentrations reasonably agree to approximately 2 g m⁻³. The model produces a higher ice number concentration for $IWC$ > 2 g m⁻³ than those observed in situ.

Figure 11b shows comparisons of the modeled mean volume diameter of ice particles with those obtained from observations. In order to match the definition of the modeled ice size, the measured mean volume diameter was calculated as $D_{3\,ice} = \left(\frac{6}{\pi \rho_{ice}} \frac{IWC}{N_{ice}}\right)^{1/3}$. As seen, the measured and modeled median $D_{3\,ice}$ agree within 25%. However, the modeled sizes have large scatter compared to the measured ones.

We emphasize that there are many limitations in this study and these clearly illustrate the need for further laboratory, numerical and field studies. On the measurement side, there are challenges with in-situ observations such as limited sampling statistics, limitations of flight operations, and instrumentation uncertainties and errors. For the numerical model, there are challenges and limitations that include unresolved turbulent air motion, hydrometeor size distribution assumptions in the bulk microphysics scheme, incomplete knowledge of natural cloud microphysics to constrain the parameterization of individual processes, and the use of a quasi-idealized model set-up. Overall, however, the GEM model simulations using the P3 microphysics scheme yields a reasonably good agreement with the in-situ observations, with the caveat that observations themselves were used to derive the FFD parameterization. This provides a strong motivation to establish better FFD formulations as a potential leading mechanism of formation of HIWC environment in tropical MCSs.





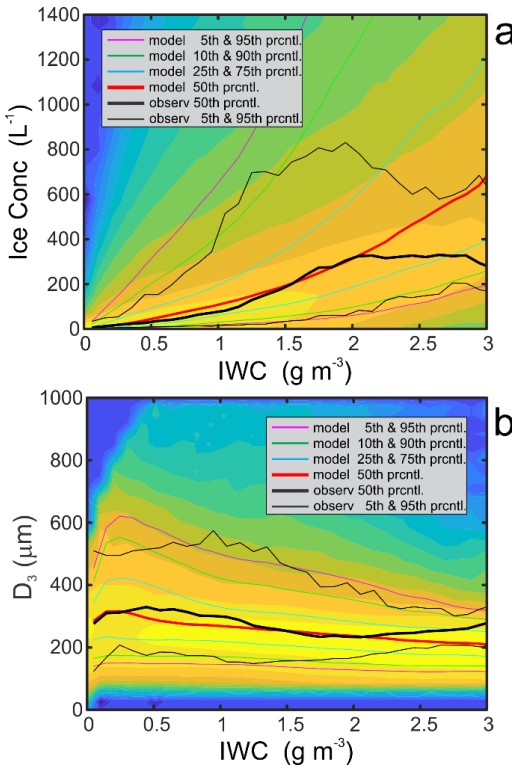

**Figure 11**. Comparisons of modeling results and in situ observations of $IWC$, $D_{3\,ice}$ and $N_{ice}$. Fields of modeled (a) concentration of ice particles >40μm vs IWC and (b) equivalent volume diameter of ice particles vs IWC. Observed median (thick) and $5^{th}$ and $95^{th}$ percentile (thin) values are shown in black. The modeling data were averaged over time period from 75 to 135 min and the range of altitudes between 5.8 and 7.4 km corresponding to the CV580 sampling altitudes.

### 3.4  SIP initiation.

The results of the SIP-ON simulation show that the cloud environment above the melting layer is densely populated with small ice crystals (Figs. 8a,c and 9c,d). Thus, within 1 km above the melting layer the average $N_{ice}$ in the SIP-ON run exceeds that of BASE by two orders of magnitude (Fig. 9c). Since the inclusion of the FFD and HM processes in the SIP-ON simulation is the only difference between the SIP-ON and BASE configurations, the enhanced concentration of ice particles above the melting layer in the SIP-ON simulation is unambiguously attributed to the secondary ice processes.

Of the two additional processes included in the SIP-ON simulation, the Hallett-Mossop (HM) process was found to be significantly less efficient in producing secondary ice compared to FFD (Fig. 8g). Therefore, the main contribution to the ice concentration is from the FFD. Equation 1 shows a strong dependence on the drop size, which suggests that the FFD rate will be most efficient in presence of large drops. This raises the question of the source of large drops inside deep glaciated MCSs.

The set of diagrams in Fig. 12 presents vertical $6 \times 8$ km fields of the main microphysical fields of interest intersecting a convective updraft in the vicinity of the melting layer. As seen from Fig. 12a, the vertical updraft starts at approximately 3 km and extends above 7 km. The convective updraft results in supersaturation over water (Fig. 12n) and activation of cloud



droplets (Fig. 12e,h,k). The boundary of the water supersaturation region (Fig.12n) and newly formed cloud (Fig. 12e,h,k) follows the pattern of the convective plume (Fig. 12a).

The Doppler velocity field in Fig. 12b shows that the majority of cloud particles above the melting layer are moved upward within this convective updraft. The depression of 5 m s$^{-1}$ of the Doppler velocity field with respect to the neighboring region below the melting layer indicates that precipitation size drops with $D$ > 2mm are lifted by the updraft. Smaller-size drops will have higher upward velocity in the updraft, and they will be brought up by the updraft to higher

altitudes during the same time period. This is consistent with the $D_{3\,rain}$ field (Fig. 12i), which shows that 1 mm drops did not ascend through the melting layer because of the slow vertical velocity of their ascent. However, drops with $D$ < 500 µm have small enough fall velocity to be moved by the convective updraft above the melting layer.

        Figure 12l shows how a convective updraft affects the field of the concentration of raindrops

$N_{rain}$. It is worth noting that the top of the melting layer is densely populated with a large number of small-sized rain and drizzle drops, the concentration of which may reach several hundred per liter. The small sizes of these small drops explained by the melting of small secondary ice particles, which melt rapidly below the freezing level. The concentration of raindrops in the melting layer of BASE is lower, because of the lower ice concentration, and the

size of raindrops is larger because of larger ice particles compared to the SIP-ON simulation.

        The upward transport of rainwater through the melting layer (Figs. 12a,f) eventually manifests in the disturbance of the bright band (Fig. 12c). Similar disturbances of the bright band are observed in situ (Figs. 6a and 7a). Note that the patterns of the fields of $LWC_{rain}$, $N_{rain}$, $D_{3\,rain}$ do not coincide with that of $LWC_{cloud}$, $N_{cloud}$, $D_{3\,cloud}$; this suggests that the rain

water is primarily transported by the updraft from underneath of the melting layer, rather than being formed locally though the collision-coalescence process of cloud water. However, accretion of cloud water may contribute to the enhancement of raindrop sizes.

        The high value of $dN_{ice}/dt$ results in the increase of $N_{ice}$ (Fig. 12j; location indicated by the arrow). As expected, the newly formed particles in the region with enhanced concentration

have smaller sizes (Fig. 12g, indicated by arrow) due to their smaller age.

        Figure 12m shows that the field of the high SIP rate $dN_{ice}/dt$ (peak location indicated by the arrow) occurs inside the fields of $LWC_{rain}$, $N_{rain}$, $D_{3\,rain}$ in Figs. 12f,i,l, but it does not necessarily coincides with their peak values. The areas maximum of $dN_{ice}/dt$, indicated by rectangles in Figs. 12f,i,l, are offset from the local maxima of $LWC_{rain}$, $N_{rain}$, and $D_{3\,rain}$. One

of the reasons for that is that besides the droplet size, the FFD rate also depends on temperature, the concentration of large drops, the ice particles sizes, and their concentrations. Consequently, combinations of these five fields may result in local maxima of $dN_{ice}/dt$, which are not necessarily coincident with the individual maxima of $N_{rain}$, $D_{3\,rain}$, $N_{ice}$, and $D_{3\,ice}$.

        Freezing of cloud and rain drops and depositional growth of numerous secondary ice

particles result in the release of latent heat, which will subsequently lead to positive temperature perturbations. The local temperature perturbation near the top of the updraft reaches approximately 3°C (Fig. 12o). Such high-temperature fluctuations will result in the invigoration of convection above the melting layer.





**Figure 12**. Model microphysical and thermodynamical fields illustrating the effects of recirculating rain drops in convective updrafts on the FFD process. (a) vertical velocity $U_z$; (b) Doppler velocity $U_{Doppl}$; (c) radar reflectivity $Z$; (d) $IWC$; (e) $LWC_{cloud}$; (f) $LWC_{rain}$; (g) mean mass diameter of ice $D_{3\,ice}$; (h) mean mass diameter of cloud droplets $D_{3\,cloud}$; (i) mean mass diameter of rain drops $D_{3\,rain}$; (j) ice particles number concentration $N_{ice}$; (k) cloud droplets number concentration $N_{cloud}$; (l) rain drops number concentration $N_{rain}$; (m) SIP rate $dN_{ice}/dt$; (n) relative humidity over liquid $RH_w$; (o) temperature perturbation $\Delta T$. The rectangles in (f), (i), (l) indicate locations of the active SIP regions shown in (m).



The analysis of the numerous cases similar to that in Fig. 12 across the model domain suggests that convective updrafts play a key role in transporting precipitation size drops across

the melting layer. The presence of large drops provides one of the necessary conditions for initiating the FFD process. However, just the presence of the large drops is not sufficient for triggering the FFD chain reaction; the four other criteria required for efficient SIP production, i.e. (1) drop concentration, (2) size of ice particles, (4) concentration of ice particles and (4) temperature must be also met. The optimum conditions for the FFD $dN_{ice}/dt$ may not

necessarily occur inside the updrafts intersecting the melting layer, but rather they may happen somewhere in their vicinity of the convection, when the outflow of large drops mixes with the pre-existing ice outside of the convective plume.

### 3.5 Spatial and temporal characteristics of SIP regions

One of the important aspects of HIWC formation is understanding where, when, and at what rate ice initiation occurs. This section analyzes the rates of ice production and spatial distributions of SIP regions in the model domain. Given that the FFD SIP process is the main contributor to ice production, for the sake of conciseness, the following analysis is focused on the FFD regions, and the discussion of the role of the HM is omitted.

Figures 13a,b show a horizontal and vertical cross sections of the FFD rate fields inside the modeled MCS. The horizontal cross section in Fig. 13a refers to the altitude of 5.8 km (-5°C) indicated by two arrows in Fig. 13b, and the vertical section in Fig. 13b was done along the dashed line in Fig. 13a. As seen from these diagrams $dN_{ice}/dt$ varies within several orders of magnitude and may reach $10^7$ m$^{-3}$ s$^{-1}$.


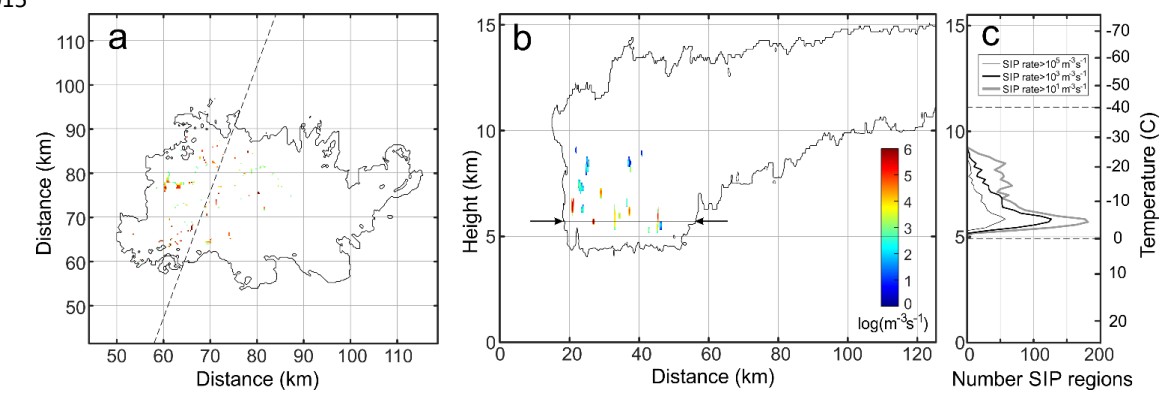

**Figure 13**. Cloud regions with active SIP simulated at 90 min. (a) Horizontal cross section of the SIP rate field at $H$ = 5.8 km, $T$ = -5°C. (b) Vertical section of SIP rate field along the line indicated by the dashed line in (a). Arrows in (b)

indicate the level of the horizontal section shown in (a). Thin black lines in (a, b) indicate cloud boundaries at the isoline of ice concentration at the threshold $N_{ice}$ = 0.01 L$^{-1}$. (c) vertical profiles of the number of spatially isolated SIP regions at different SIP rate thresholds for the cloud shown in (a) and (b).

Both panels in Fig. 13 show that regions of secondary ice production have highly clustered

spatial structures. The vertical changes of the number of the isolated SIP regions ($n_{SIP}$) integrated over each model level in Figs. 13a,b is shown in Fig. 13c. As seen from Fig. 13c $n_{SIP}$ depends on the threshold of $dN_{ice}/dt$. The general trend of the vertical changes of $n_{SIP}$ for all three SIP thresholds remains the same and they all reach a maximum at the same altitude ~5.8 km (-5°C).



The production of secondary ice extends up to 9.2 km, which corresponds to -25°C, the lowest
temperature of the FFD process (Keinert et al., 2020).

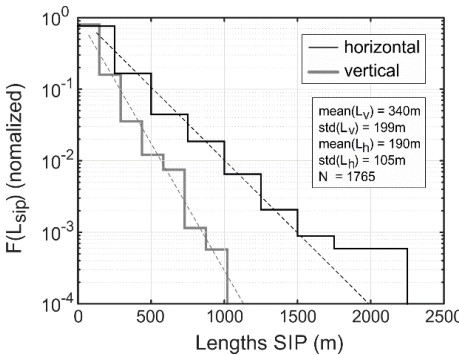

**Figure 14**. Frequency distributions of vertical $L_v$ and horizontal $L_h$ lengths of SIP clouds regions with SIP rate >$10^3$ m$^{-3}$ s$^{-1}$. The spatial averaging was performed over the in-cloud layer 5.0<H<9.2 km at 90 minutes of modeled cloud. Dashed lined are parameterization of the frequency distributions.

Figure 14 shows frequency distributions of horizontal $F(L_h)$ and vertical $F(L_v)$ lengths of the SIP regions. These distributions were integrated over the cloud layer between 5.2 km and 9.2 km (30 model levels). The mean values of $L_h$ and $L_v$ are equal to 340 and 190 m, respectively. However, in terms of the modeled grid cell dimensions (i.e., horizontal = 250 m and vertical ≈ 145 m) the mean values of $L_h$ and $L_v$ are 1.36 and 1.31, respectively. That is, in the modeled space the shapes of SIP regions are isometric, with $L_h \approx L_v$. This suggests that the resolution of the model may not be sufficient to resolve the actual dimensions of the SIP regions and that SIP is an essentially a subgrid process (within the context of a model with a 250-m horizontal grid spacing).

Both $F(L_h)$ and $F(L_v)$ can be well parameterized by an exponential function $F(L) = ae^{-bL}$ where $a_v$= -8.2 $10^{-3}$; $b_v$= 0.024; $a_h$= -4.7 $10^{-3}$; $b_h$= 0.089. As follows from Fig. 14, approximately 75% of active SIP regions $L_h$ and $L_v$ have scales of approximately one grid cell, i.e. 250 m and 145 m, respectively.

The $F(L_h)$ and $F(L_v)$ distributions were calculated for the SIP regions with the rate $dN_{ice}/dt$ > $10^3$ m$^{-3}$ s$^{-1}$. Calculations of $F(L_h)$ and $F(L_v)$ for other SIP threshold in the range $10^1$ m$^{-3}$ s$^{-1}$ < $dN_{ice}/dt$ < $10^5$ m$^{-3}$ s$^{-1}$ showed, that it does not have any significant effect on the shape of the distributions in Fig. 14 and the mean values of $L_h$ and $L_v$ vary within only 7% .

Another important characteristic of SIP regions is their spatial occurrence. Figure 15 shows vertical profiles of the area fraction and the average distance between SIP regions. The area fraction is defined as the ratio of the area of all SIP regions in the modeled layer to the area of the horizontal cloud cross section at that level. The cloud cross-sectional area was calculated for the environment with $N_{ice}$ > 0.01 L$^{-1}$ (e.g., Fig. 13a). As seen from Fig. 15a, the SIP area fraction varies between 0.2% and 2% depending on the SIP rate threshold, and its maximum occurs between 5.5 and 5.8 km. This altitude corresponds to the maximum rate of SIP (Fig. 8g) and maximum number of SIP regions (Fig. 13c).

The average distance between SIP regions, depending on the SIP threshold, varies from approximately 30 to 100 km at the level of maximum SIP occurrence (5.5 to 5.8 km). Above and below this altitude the average distance between SIP regions rapidly increases.



Analysis of the temporal evolution of SIP regions is quite challenging. However, visual analysis of the cross section of the domains of $dN_{ice}/dt$ allowed a rough assessment that the lifetime the SIP regions is limited by 2-3 min.

The primary findings of this section can be summarized as follows. The FFD SIP regions are short-living cloud objects (~$10^2$ s) with spatial scales of the order of hundreds of meters. The area fraction of SIP regions is quite small and typically less than 1%. At the level of their maximum SIP occurrence, the average distance between the SIP regions is of the order of tens of kilometers. Altogether, the spatial sparsity and short lifetime make the SIP regions challenging objects for in-situ observations.

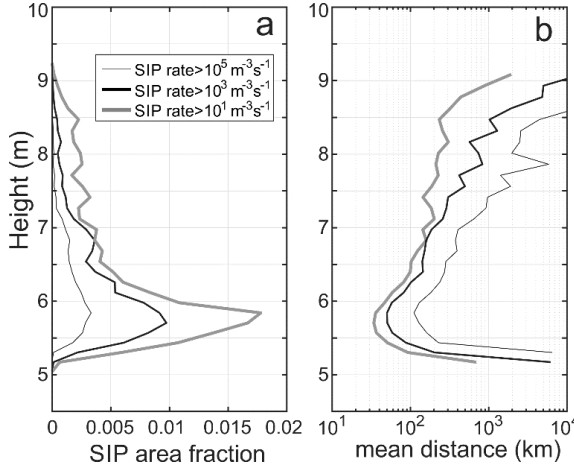

**Figure 15**. Vertical profiles of (a) the area fraction and (b) mean distance between the SIP cloud regions.

Existing observational techniques do not allow for measurements of SIP rates. However, the spatial scales of the cloud regions that experienced SIP in the recent past could be estimated from the measurements of horizontal lengths of cloud regions with enhanced concentrations of small facetted ice crystals ($D_{max}$<100 μm). Such small ice crystals, due to their small age, are still associated with the location of their origin, and the dimensions of the cloud regions affected by SIP have not yet been significantly changed by turbulent diffusion. Therefore, the dimensions of the cloud regions with enhanced concentrations of small ice particles can be used as a proxy of the active SIP regions. Thus, Korolev et al. (2020 Figs. 5a, 23a) showed that typical horizontal dimensions of such cloud regions are of the order of a few hundred meters. The same study showed that SIP regions are separated by tens of kilometers. Therefore, spatial scales, clustering of SIP cloud regions, and distance between them obtained in simulations are generally consistent with the in-situ observations.

## 4. Summary of observations and modeling

Prior to the formulation of the conceptual model of HIWC, we summarize the main outcomes of in-situ observations (section 2) and numerical simulation (section 3), which are relevant to the understanding of the formation of HIWC:

(1) Small facetted hexagonal prisms ($D_{max}$<100μm) with measured number concentrations up to $N_{ice}$~$10^3$ L$^{-1}$ were observed above the melting layer at temperatures $T < -2°C$. The




enhanced concentrations of small ice particles were attributed to the massive production
of secondary ice (section 2.2).

(2) The secondary ice production cloud regions with enhanced concentration of small
facetted ice were frequently accompanied by observations of supercooled large drops
($D < 300\mu m$), deformed frozen drops and fragments of frozen drops. This enables the
hypothesis that the main mechanism of SIP is the fragmentation of freezing drops (FFD).
(section 2.2).

(3) Cloud regions experiencing SIP were sparce and had small horizontal extensions (i.e., of
the order of $10^2$ m). SIP regions were typically associate with convective regions (section
2.2).

(4) The dominating ice shapes in tropical MCSs has a distinct pattern in the vertical direction
above the melting layer: "small hexagonal prisms" => "columns" => "capped columns".
Such vertical changes of the ice particle habits can be explained by metamorphosis of ice
particles when they experience successive columnar growth (-9°C < $T$ < -2°C) followed by
plate growth (-22°C < $T$ < -9°C) regimes. The observed changes of ice particles in the
vertical direction suggest the small ice particles formed in the vicinity of the melting layer
were transported upwards by the convective updrafts and went through columnar and
then plate growth regimes. (section 2.3).

(5) The HIWC cloud regions on average have significantly higher concentrations and smaller
ice particles compared to the cloud regions with low IWC. (section 2.3; Figs. 3,4; Table 1)

(6) Mixed-phase cloud regions in mature MCSs are sparce and occupy only a few percent of
in-cloud space at -5°C. Mixed-phase cloud regions were primarily observed in convective
updrafts which can force coexistence of ice and liquid phase (section 2.5).

(7) Spatially sparce large drops (100 $\mu m$ < $D$ < 300 $\mu m$) with number concentration < $10^{-3}$ L$^{-1}$
were observed at 1-2 km above the melting layer in the vicinity of updrafts. In absence of
any significant LWC layers the only source of these large drops could be precipitating size
drops formed below the melting layer and then transported upward by convective
updrafts (section 2.6).

(8) The disturbance of the bright band was typically observed in convective regions and is
explained by vertical transport of melting ice along with precipitation size drops through
the melting layer in an upward direction (section 2.8).

(9) HIWC cells are associated with convective updrafts existing somewhere inside these cells
(section 2.9).

The numerical model simulation (SIP-ON configuration) reproduced the observed large
concentration of ice particles above the melting layer, with an average ranging from $10^2$ L$^{-1}$ to
$10^3$ L$^{-1}$ and peaking up to $10^4$ L$^{-1}$. The main mechanism of ice initiation was related to the FFD
process.

The model was able to reproduce high clustering and sparsity of SIP regions when using a
parameterization that was based on previous aircraft measurements. The typical horizontal
dimensions ($10^2$-$10^3$ m) and distances between SIP regions ($10^1$-$10^2$ km) are generally
consistent with observations. The model qualitatively and quantitatively reproduced trends of
vertical profiles of $IWC$, $LWC$, $N_{ice}$, $D_{3\,ice}$, $Z$, and $U_{Doppl}$ (Fig. 10) and general relations
between $N_{ice}$ and $D_{3\,ice}$ versus $IWC$ (Fig. 11).

The numerical modeling confirmed the hypothesis of transportation of precipitation size
drops through the melting layer by convective updrafts (Korolev et al., 2020). The model also



reproduced the disturbance of the bright band and justified the hypothesis of disturbance of the bright band by convective updrafts. The vertical transport of precipitation size drops creates an environment above the melting layer favorable for the SIP FFD process.

The agreement between in-situ observations and numerical simulations creates a basis for the following conceptual model of the formation of HIWC cloud regions in tropical MCSs.

## 5.    Conceptual model of HIWC

The combination of in-situ observations and numerical simulations enables us to make the following general conclusions regarding formation of HIWC in oceanic tropical MCSs. The
process begins with forming a melting layer in a convective environment during the initial stage of MCS. Updrafts generated by regular convection or gravity waves transport precipitation sized drops from underneath the melting layer to above it. Above the melting layer, large supercooled drops create a favorable environment for the FFD SIP process. FFD yields a massive production of secondary ice particles within 1 to 1.5 km above the melting layer with a
maximum of around -5°C to -8°C. Freezing drops and rapid depositional growth of large amounts of ice particles result in enhanced release of the latent heat and invigoration of the convection. During the subsequent updraft, the ensemble of small ice crystals rapidly depletes the supersaturated water vapor, increasing IWC, thereby forming the HIWC environment. High concentrations of secondary ice particles lead to intense competition for the water vapor,
which hinders their growth. As a result, ice particles constituting the HIWC environment remain small in size. The convective updrafts transport small ice particles to higher altitudes and distribute them throughout the stratiform regions.

Convective updrafts across the MCS domain periodically pump small ice crystals to high altitudes, maintaining the HIWC environment there. The maintenance of HIWC is also facilitated
by small sizes of ice particles, which may remain suspended in the cloud for a longer time due to the small fall speeds. Overall, HIWC environments are dynamic cloud objects that form due to a balance between particle sedimentation and IWC brought up by convection.

The conceptual model of the HIWC formation is illustrated in Fig. 16.

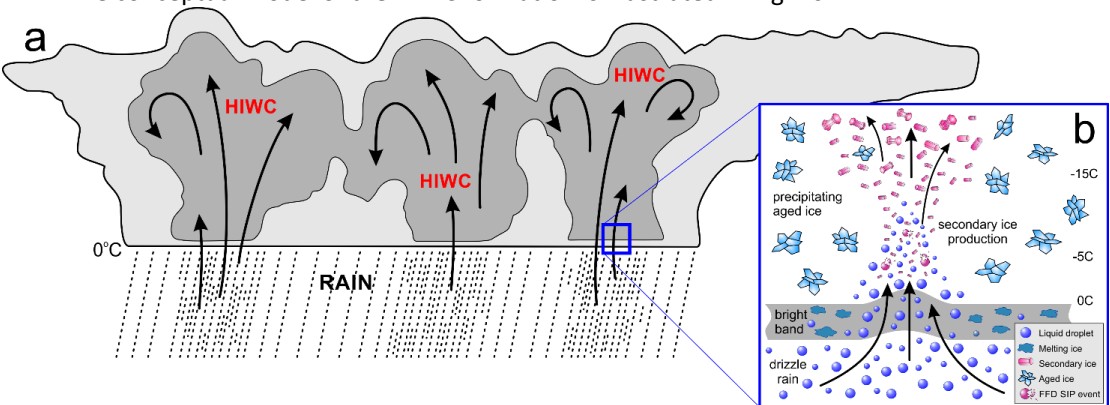

**Figure 16**. Illustration of a conceptual model of HIWC formation in tropical MCSs

While the proposed conceptual model is based on a relatively limited number of in-situ observation (focused on one specific geographical location) and quasi-idealized numerical model simulations, both of which have limitations and challenges the central point of the



conceptual model is that HIWC environments in tropical MCSs form as a result of massive
       production of secondary ice particles at temperatures just below freezing (primarily -12°C < $T$ <
       -2°C), which are then transported to high altitudes (~7 < $H$ < 15 km) by convective updrafts. This
       part of the conceptual model is well justified based on in-situ observations and can be
       considered as an established experimental fact.

1180       However, despite observational evidence indicating that the FFD process is the major
       contributor to the production of secondary ice, this part of the conceptual model requires more
       in-situ observations and especially laboratory studies. The latter is crucial for detailed studies of
       the dependence of FFD rates on environmental conditions and the development of physically-
       based parameterizations that can be used in numerical simulations. The simulations examined

in this study support the hypothesis that FFD SIP alone is sufficient to explain the large amounts
       of ice generated without involving other SIP mechanisms (Korolev and Leisner, 2020; James et
       al., 2021). That said, the authors acknowledge that the FFD mechanism is not well established
       in the laboratory, and the contribution of other mechanisms also remains uncertain and
       requires further study.


       **6.   Concluding remarks**
       The mechanism of HIWC formation described above is specifically relevant to tropical
       oceanic MCSs, which are characterized by a low aerosol load and relatively moderate
       convection with vertical velocities peaking at 15 to 20 m s-1. We emphasize that the roles and

contributions of different cloud processes in HIWC depend on the initial environmental
       parameters, and it may be different for continental and midlatitude convective systems
       compared to HIWC in maritime tropical MCSs. Results of the baseline simulation, with SIP
       processes off, suggest that any convective system, if strong enough, may generate a HIWC
       environment. However, the bulk properties of the population of ice particles comprising HIWC,

such as MMD, reflectivity, and IWC, may be different from those observed in the tropics. For
       example, as discussed in Lawson et al. (2015, 2023), high aerosol load in continental convective
       systems may suppress SIP. One of the consequences of SIP suppression is the high production
       of graupel/hail, increased radar reflectivity, and relatively low ice number concentration. This
       case would be equivalent to the HIWC environment in the BASE simulations as described in

section 3.2 (Figs.8). Due to the larger particle sizes and, therefore, higher fall velocity, the
       endurance of the HIWC environment in such clouds will be shorter compared to those in
       tropical MCSs with smaller ice particles.
           We reiterate that while the numerical simulations presented in this study support the
       proposed conceptual model for the importance of SIP in the production of HIWC in tropical

convection, no claim is made regarding the fidelity of the model in accurately simulating the
       fundamental processes involved in SIP.  Besides the model set-up being quasi-idealized, the FFD
       parameterization itself in the microphysics scheme is ad hoc and is essentially constructed to
       give good results.  The original FFD formulation in Lawson et al. (2015) is essentially an
       empirical parameterization based on observations; in this study we have adjusted it further to

produce peak FFD rates at relatively warm temperatures (in contrast to laboratory studies) in
       order to improve the simulations.  The utility of this model "tuning" approach is that while we
       have insufficient knowledge of the fundamental physics on which a credible FFD
       parameterization can be based, we can at least introduce what may be considered a



compensating model error such that the end result (i.e. the numerical simulation with SIP on) is sufficiently close to the in-situ microphysical observations that we can use the simulation to examine further the model storm and to lend support to the largely observation-based conceptual model.

The significance of the initial environmental conditions for the bulk properties of the HIWC environment was supported by the recent in-situ observations of the HIWC cloud regions in
cold season mid-latitude frontal clouds (Rugg et al., 2022). Despite the small data set, it was found that in the mid-latitude winter clouds, MMD in HIWC regions (Fig. 10, in Rugg et al., 2022) was higher compared to those in tropical MCSs (Fig. 17, in Strapp et al., 2021). These comparisons indicate the need for more observational and modeling studies of bulk properties of HIWC cloud regions in different geographical regions with different environmental
backgrounds. An especially high priority should be placed on furthering reproducible laboratory-based parameterizations of FFD and other SIP mechanisms.

*Data availability*: In situ data are available from the Earth Observation Laboratory (EOL) archive https://doi.org/10.26023/PSC2-TTQS-390A. The GEM code model is available at
https://github.com/ECCC-ASTD-MRD/gem. The code for the P3 microphysics scheme used is available at https://github.com/P3-microphysics/P3-microphysics). Configuration files to reproduce the GEM simulations are available upon request.

*Authors' contribution.* A. Korolev developed a concept, led the collection of the cloud microphysical data
and data analysis. Z. Qu performed GEM model simulations. J. Milbrandt is the co-developer of the P3 microphysics scheme. I. Heckman performed analysis of the cloud microphysical data. M. Wolde carried out the airborne data collection and analysis of in-situ atmospheric state parameters and NAWX radar data. N. Cung performed analysis on the NAWX radar data. M. Cholette developed and implemented the predicted liquid fraction capability of the P3 scheme and leads its maintenance the GEM model. G.
McFarquhar, A. Fridlind participated in planning of HIWC flight operations during HAIC-HIWC 2015. A. Korolev prepared the manuscript, with contribution from co-authors G. McFarquhar, P. Lawson, A. Fridlind, Z. Qu, and J. Milbrandt.

*Competing interests*. The authors declare that they have no conflict of interests. At least one of the co-
authors is a member of the editorial board of Atmospheric Chemistry and Physics.

*Acknowledgements.* The HIWC program was supported by Environment and Climate Change Canada (ECCC), the National Research Council (NRC), Transport Canada (TC), and the Federal Aviation Administration (FAA). Any opinions, findings, conclusions, or recommendations expressed in this
material are those of the authors and do not necessarily reflect the views of the FAA and TC. Authors express special thanks to the NRC FRL pilots Anthony Brown, Paul Kissman, and Rob Erdos for their outstanding cooperation during flight operations and in-situ data collection. The authors are grateful for the technical support provided by the ECCC and NRC technical and engineering teams. Authors acknowledge contributions from Ed Emery, Tom Ratvasky (NASA GRC), Walter Strapp (MetAnalytics
Inc.), Lyle Lilie (SAE Inc.).



**APPENDIX A**

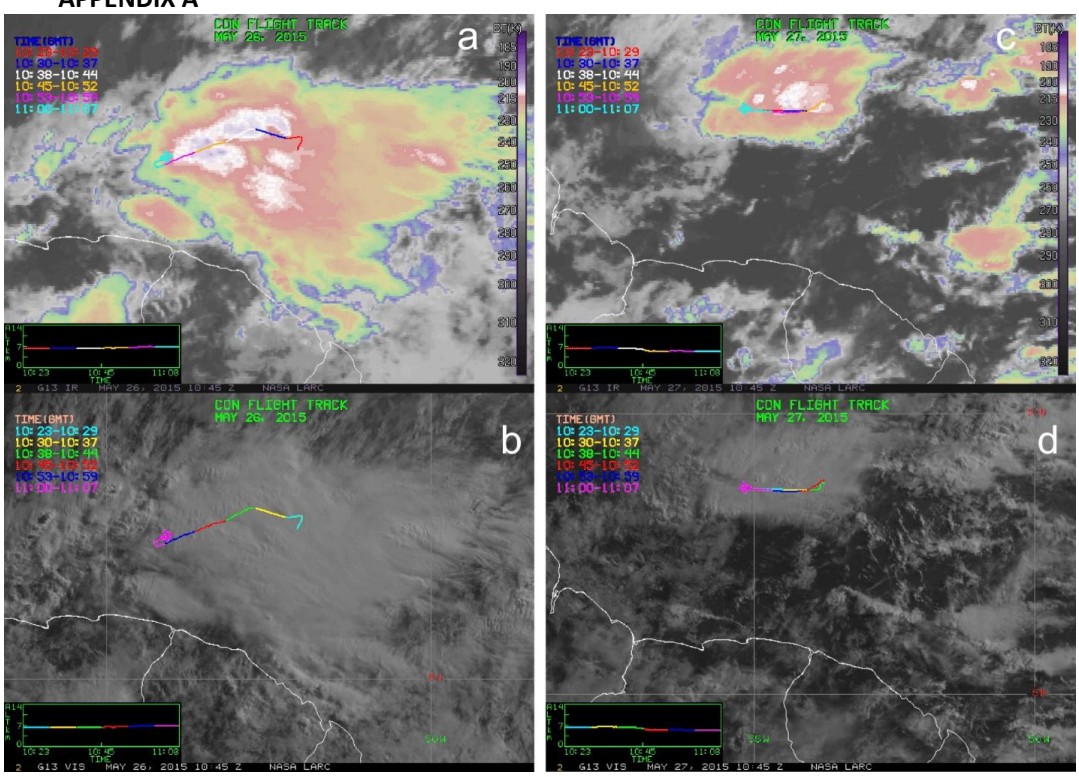

**Figure A1**. GOES-13 satellite infrared and visible images with the Convair-580 flight track for the flight segments 1265 shown in (a,b) Fig. 6 and (c,d) Fig. 7.







**APPENDIX B**

**Table B1: Summary of GEM configurations details.**

| Dynamics/Numerics |
| --- |
| • Nonhydrostatic primitive equations |
| • Limited-area grid on a latitude–longitude projection |
| • Uniform horizontal grid spacing of 0.00225 longitude (~ 250 m) |
| • 183 vertical levels |
| • Upper-boundary nesting above 10 hPa |
| • Time step of 10 s |
| • Terrain-following Gal-Chen vertical coordinate |
| • Two-time-level semi-implicit time differencing |
| • 3D semi-Lagrangian advection |
| • $\nabla^4$ horizontal diffusion ($\nabla^6$ for potential temperature) |
| **Physics** |
| • Planetary boundary layer scheme based on turbulence kinetic energy with statistical representation of subgrid-scale cloudiness (MoisTKE) |
| • Kuo–transient shallow convection scheme |
| • P3 microphysics (with triple-moment-ice and liquid fraction options on) |
| • Li–Barker correlated-k distribution radiative transfer scheme (called every 3 min) |
| • Interaction Sol-Biosphère-Atmosphère (ISBA) land surface scheme |
| • Distinct roughness lengths for momentum and heat/humidity |


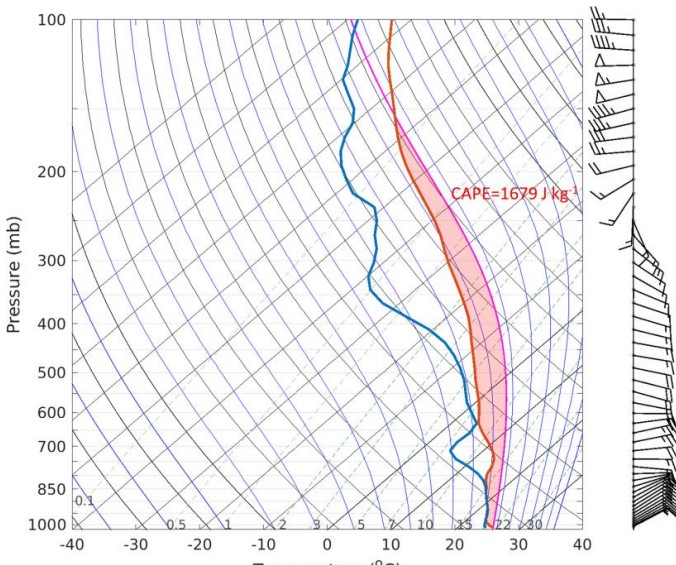

**Figure B1.** Initial atmospheric profiles for the idealized simulations. The blue line represents the dew point temperature, the red line represents the environment sounding (temperature), and the magenta line represents the parcel lapse rate. (Qu et al., 2022)




**APPENDIX C**

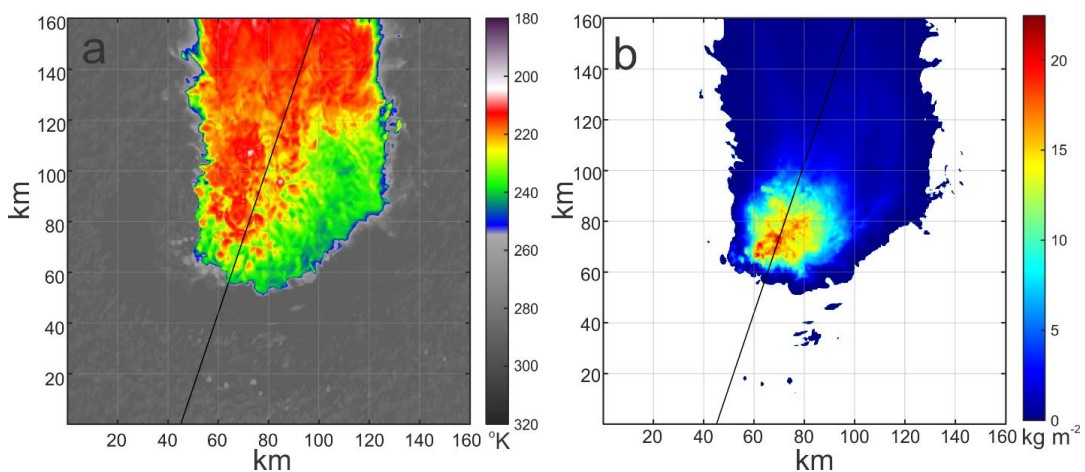

**Figure C1**. Brightness temperature (a) and ice water path (b) of the GEM simulated cloud system at 90 min. Black lines indicate the cross sections of the microphysical fields shown in Fig. 8.


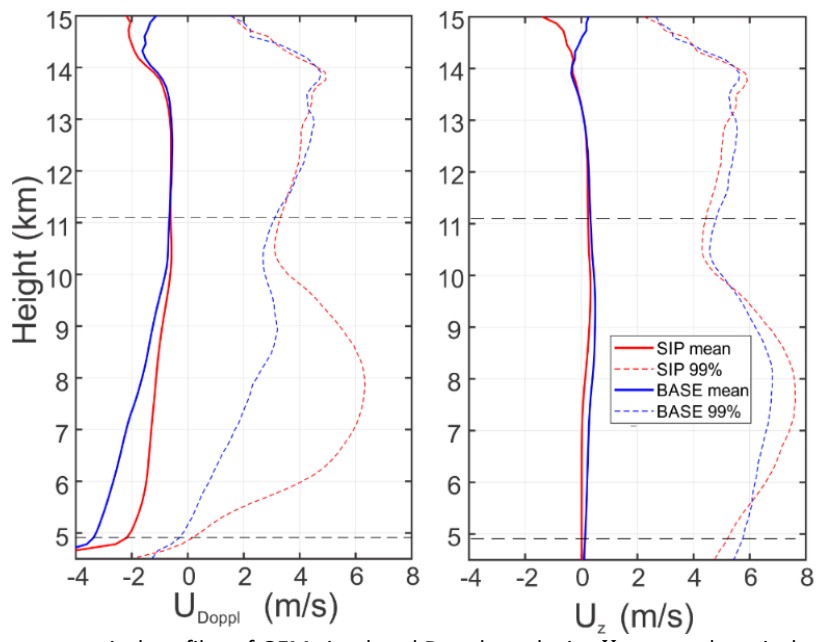

**Figure C2**. Average vertical profiles of GEM simulated Doppler velocity $U_{Doppl}$ and vertical wind velocity $U_z$. The averaging was performed over time period from 75 to 135 min.



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
