# Peer review of "High ice water content in tropical mesoscale convective systems (a conceptual model)"

_EGUsphere, 2024_

## Referee Comment (RC2)

[referee-annotated manuscript omitted]

---

## Author Comment (AC1)

**Replies to the comments of Reviewer 2**

The authors appreciate the thoughtful and helpful comments of Reviewer 2 (Andy Heymsfield). Below are the authors' point-by-point replies, which are highlighted in blue. The comments below appear following the sequence of original comments in the pdf file.

1. *Comment*: Ryan
*Reply*: The reference Ryan et al. (1976) was added in the manuscript: Ryan, B. F., Wishart, E. R., and Shaw, D. E.: The Growth Rates and Densities of Ice Crystals between −3°C and −21°C. J. Atmos. Sci., 33, 842-850, https://doi.org/10.1175/1520-0469(1976)033<0842:TGRADO>2.0.CO;2, 1976.

2. *Comment*: 282-283. Reference
*Reply*: References on the studies on the dependence of ice particle habits of temperature were added:
   Magono, C. and C. Lee, Meteorological classification of natural snow crystals, J. Fac. Sci., Hokkaido Univ. Ser. VII, *2*, 321-335, http://hdl.handle.net/2115/8672, 1966.

   Kobayashi, T.: The growth of snow crystals at low supersaturation. Phil. Mag., 5, 1363-1370, https://doi.org/10.1080/14786436108241231, 1961.

   Rottner, D., and Vali, G.: Snow crystal habit at small excesses of vapor density over ice saturation. J. Atmos. Sci., 31, 560–569, https://doi.org/10.1175/1520-0469(1974)031,0560:SCHASE.2.0.CO;2, 1974.

3. *Comment*: 294. Further up>Higher
*Reply*: Corrected as per Reviewer's comment.

4. *Comment*: 354. "skipped"to omitted
*Reply*: Corrected as per Reviewer's comment.

5. *Comment*: 364-366. This could also be done using the IKP and assuming that the RH is 100 wrt water and also 100% wrt ice.
*Reply*: The first author of the paper performed an extended analysis of the assumption that RHice=100% in HIWC environment. Despite the trend of increasing mean RHice with the increase of IWC, in many cases ice clouds may be undersaturated up to 20%. At relatively high temperatures from -5C to -10C this may result in IWC biases of 0.1-0.5g/m3. The results of the study of humidity in the HIWC environment were presents at the poster session at the AMS Cloud Physics conference in 2018 in Vancouver (Korolev et al. 2018). For environments with low IWC such big errors may also lead of negative IWC when assuming 100% ice saturation. However, the assumption RHice=100% works well for temperatures T<-30C where the specific humidity at ice saturation is small and therefore, even if the relative humidity is different from saturated over ice, the error in IWC will be small (~<0.01g/m3)

   Korolev, A., Heckman, I., and Wolde, M.: Observation of Phase Composition and Humidity in: Oceanic Mesoscale Convective Systems, 15th AMS Cloud Physics Conference, Vancouver, BC, 9–13 July 2018, 1450 available at: https://ams.confex.com/ams/15CLOUD15ATRAD/webprogram/Paper347111.html (last access: 30 April 2024), 2018.

6. *Comment*: 367. What range of A and B were found?

*Reply*: In the original paper by Brown and Francis (1995), the coefficients A and B in the size-to-mass parameterization were obtained for the irregular shape ice particles. However, the HIWC environment is dominated by columnar ice crystals, and therefore, it is anticipated that the coefficients A and B will differ from those obtained in Brown and Francis (1995). Optimization of A and B based on the comparisons with the IKP yields the best results for A= 7.0044e-12 and B=2.3. A text with the above explanation was added on page 25.

7. *Comment*: 403. scatter

*Reply*: Corrected as per Reviewer's comment.

8. *Comment*: Important. What liquid water contents do you get at temperatures warmer than -2C?

*Reply*: During the HIWC project 2015, no sustainable sampling of cloud microstructure was performed at temperatures 0C to -2C. The clouds at this temperature range were sampled during transit to the area of operation, and therefore, they are not statistically representative of the cloudy environment in this temperature range.

9. *Comment*: Important. 684. Based on your earlier discussion, the FFD process would only be important at temperatures of about -2 to -8C because the particles would begin as needles/columns and then be transported to lower temperatures.

*Reply*: In this regard, it is worth noting that, at present, there is no lab data on the rate of production of ice splinters versus temperature during FFD SIP. The results of the lab studies by Keinert et al. (2020) reported the frequency of breakups, cracking, jetting, and bubble bursts during FFD. However, the rate of secondary ice production during these events remains undetermined. Under these circumstances, we employed a heuristic approach based on the FFD SIP parameterization in Lawson et al. (2015) and P3 scheme adjustment to match the observations in studied MCSs.

10. *Comment*: 728. "large" to "high"

*Reply*: Corrected as per Reviewer's comment

11. *Comment*: 738. is reverse

*Reply*: Corrected as per Reviewer's comment

12. *Comment*: 740. Important. Are the vertical velocities higher in SIP-ON than BASE?

*Reply*: The average vertical velocities in SIP-ON and BASE are approximately the same (Fig. C2 in Appendix C). However, the average Doppler velocity in SIP-ON is higher than in BASE because the larger particles in BASE simulation. This question is discussed in the text of the paper

13. *Comment*: 828. purpose

*Reply*: Corrected as per Reviewer's comment.

14. *Comment*: 880. An incomplete gamma distribution with a relationship between the IWC and maximum dimension would help this problem

*Reply*: This is a very good suggestion. We will apply it in our future simulations.

15. *Comment*: 886-888. For X-band and graupel, non-Rayleigh effects are probably small
*Reply*: Authors agree with this statement.

16. *Comment*: 891. How are the particle terminal velocities calculated?
*Reply*: The following text has been added on page 19 to address the reviewer's comment: "The Doppler velocities were calculated offline by subtracting the mass-weighted fall speed of hydrometeors from the vertical wind speed. In the P3 scheme, the mass-weighted fall speed for each ice category is computed based on the fall speed for an individual particle of a given size and set of bulk predicted bulk properties (following Heymsfield et al., 2007 and integrated over the particle size distribution (see Morrison and Milbrandt, 2015 for details). The overall mass-weighted fall speed is then obtained by summing the bulk fall speeds of all four ice categories and for rain, weighted by their respective masses."

17. *Comment*: 1103 widths or extents
*Reply*: Corrected as per Reviewer's comment.

18. *Comment*: 1107 "small hexagonal prisms, needles
*Reply*: Needles were observed only occasionally in the studied tropical MCSs. The dominant ice particle habit was "column". Therefore, the authors prefer keeping the text unchanged.

19. *Comment*: 1139 transport
*Reply*: Corrected as per Reviewer's comment.

20. *Comment*: 1150 "initial" to "early"
*Reply*: Corrected as per Reviewer's comment.

21. *Comment*: 1152 below
*Reply*: Corrected as per Reviewer's comment.

22. *Comment*: Important. After the liquid water is depleted, doesn't SIP stop and therefore HIWC regions largely disappear. Lifetime of HIWC regions?
*Reply*: Thank you for the comment. The presence of liquid drops is one of the necessary conditions for the FFD SIP. After their depletion (e.g., due to freezing, evaporation due to WBF process, or fallout) the FFD SIP process stops. The lifetime of the SIP cloud regions is discussed in section 3.5 (page 32) We also added in Supplementary Materials an animation of SIP regions with the SIP rate $dN_{ice}/dt$ > $10^5$ m$^{-3}$ s$^{-1}$ to address the reviewer's comment. These animations allow visual assessment of the endurance of SIP cloud regions.

---

## Author Comment (AC2)

**Replies to the comments of Reviewer 1**

Authors appreciate thoughtful comments of Reviewer 1.
Below are the authors' point-by-point replies highlighted in blue.

1. *Comment*: An excellently prepared manuscript in all regards. Excellently presented, well argued, balanced etc. etc.
*Reply*: Authors thank the overall assessment of our work.

2. *Comment*: Line 110: Typo (repetition): "It was found that the that the....."
*Reply*: Corrected as per Reviewer's comment.

3. *Comment*: Line 209: "These were cleared with special image processing algorithms." Some minor elaboration on what additional data filtering was applied filtering would be useful (but not essential).
*Reply*: Given the absence of consensus on 2D data processing in the cloud physics community and a multitude of different processing algorithms, this is a valid question. However, the authors consider that minor elaboration would not be sufficient to address this comment, and its inclusion will be distractive and fuzzy. On the other hand, a detailed explanation of this question deserves a separate paper, and the inclusion of a lengthy explanation in this manuscript in the main text or appendix will unreasonably blow up the size of the paper. Therefore, for the sake of conciseness, the authors prefer to keep the related text without changes. Along this way, four co-authors of this paper are working on the manuscript summarizing the 2D data processing techniques, where this question is elaborated in detail.

4. *Comment*: Figure 1: The images in panel a (top) are an important part of the narrative, but are hard to see. Please make them bigger.
*Reply*: A high resolution zoomed-in image of cloud particles shown in Fig.1a was put in Appendix A to address the Reviewer's comment

5. *Comment*: Line 520/Figure 6: The details of the BB are hard to see. Maybe add some lines to indicate the times of interest relating to the BB disturbances. Also, could the y-axes be zoomed in to the region of interest -Is the radar profile below 3km needed?
*Reply*: To address the Reviewer's comment, three arrows were added on the top of the figure, indicating the area of interest with elevated bright bands.

6. *Comment*: The authors try to avoid making strong statements about the exact SIP process at play, but there is a strong sense of preference towards FFD. Given that both FFD and HM processes are very uncertain in terms of process rates, HM could be dominant still.
*Reply*: In the absence of firm laboratory studies of rates of secondary ice production during FFD and HM processes confirmed by independent research groups, the authors prefer to avoid strong statements about the mechanisms of observed in-situ secondary ice production. Recent lab studies of the HM SIP by Siedel et al. 2024 (https://doi.org/10.5194/acp-24-5247-2024) suggested that the rate of this process in past lab experiments was overestimated. However, more studies are still required before reaching a consensus about the significance of the HM mechanism.